

**Contrasting seasonal changes in total and intense precipitation in the**
**European Alps from 1903 to 2010**
Ménégoz, Martin[1], Valla, Evgenia[1], Jourdain, Nicolas. C.[1], Blanchet, Juliette[1], Beaumet, Julien[1],
Wilhelm, Bruno[1], Gallée, Hubert[1], Fettweis, Xavier[2], Morin, Samuel[3], Anquetin, Sandrine[1]
1 CNRS, Université Grenoble Alpes, Institut de Géosciences de l'Environnement (IGE), 38000 Grenoble, France
2 F.R.S.-FNRS, Laboratory of Climatology, Department of Geography, University of Liège, 4000 Liège, Belgium
3 Univ. Grenoble Alpes, Université de Toulouse, Météo-France, CNRS, CNRM, Centre d'Études de la Neige, 38000
Grenoble, France
**Abstract**
Changes of precipitation over the European Alps are investigated with the regional climate
model MAR applied with a 7-km resolution over the period 1903-2010 using the reanalysis
ERA-20C as forcing. A comparison with several observational datasets demonstrates that the
model is able to reproduce the climatology as well as both the inter-annual variability and the
seasonal cycle of precipitation over the European Alps. The relatively high resolution allows to
estimate precipitation at high elevations. The vertical gradient of precipitation simulated by
MAR over the European Alps reaches 33% km-1 (1.21 mm.day-1.km-1) in summer and 38%
km-1 (1.15 mm.day-1.km-1) in winter, on average over 1971-2008 and shows a large spatial
variability. A significant (p-value<0.05) increase in mean winter precipitation is simulated in the
North Western Alps over 1903-2010, with changes typically reaching 20 to 40% per century.
This increase is mainly explained by a stronger simple daily intensity index (SDII) and is
associated with less frequent but longer wet spells. A general drying is found in summer over the
same period, exceeding 20 to 30% per century in the Western plains and 40 to 50% per century
in the Southern plains surrounding the Alps, but remaining much smaller (<10%) and not
significant above 1500 m.asl. Below this level, the summer drying is explained by a reduction of
the number of wet days, reaching 20% per century over the Northwestern part of the Alps and
30-50% per century in the Southern part of the Alps. It is associated to shorter although more
frequent wet spells. Maximum daily precipitation index (Rx1day) takes its highest values in
autumn in both the Western and the Eastern parts of the Southern Alps, locally reaching 50 to 70
mm.day-1 on average over 1903-2010. Centennial maxima up to 250 to 300 mm.day-1 are
simulated in the Southern Alps, in France and Italy, as well as in the Ticino valley in
Switzerland. Over 1903-2010, seasonal Rx1day shows a general and significant increase at the
annual timescale and also during the four seasons, reaching local values between 20% and 40%
per century over large parts of the Alps and the Apennines. Trends of Rx1day are significant (p-
value<0.05) only when considering long time series, typically 50 to 80 years depending on the
area considered. Some of these trends are nonetheless significant when computed over 1970-
2010, suggesting a recent acceleration of the increase in extreme precipitation, whereas earlier
periods with strong precipitation also occurred, in particular during the 1950s/1960s.





## 1. Introduction

The European Alps are often considered as the "water tower" of continental Europe (Beniston et al., 2018), hosting the headwaters of several major European rivers, such as the Rhine, the Danube, the Po, and the Rhône Rivers. Similar to other mountain regions of the world, this stems from both enhanced precipitation rates compared to surrounding lowlands, and by the specific role played by glaciers and the mountain snow cover in regulating the local and regional hydrological cycle (Wanner et al., 1997, Viviroli et al., 2008; 2019). The high population density and the presence of strong slopes makes these areas particularly prone to natural hazards such as landslides, floods and avalanches, which are strongly connected to meteorological conditions (e.g. Beniston M., 2006; Evin et al., 2019). There is widespread evidence from scientific studies addressing past and future climate conditions, that significant atmospheric warming, largely due to anthropogenic forcing (Hock et al., in press), has been and is projected to occur in the European Alps. This warming is estimated to 1.2°C per century over the 20th century (Auer et al., 2007), which is twice as large as global rates (Brunetti et al., 2009; Gobiet et al., 2014). Precipitation changes are more difficult to detect because of (i) the difficulty to observe and simulate this variable over mountainous areas, (ii) the stronger influence of internal variability and (iii) the spatial heterogeneity peculiar to complex topography areas.

The longest time series of observed precipitation in the European Alps are available in Switzerland and Italy, with station data that have been homogenized over the last 100 to 150 years (i.e starting from 1900 and even before; Schmidli et al., 2002; Brugnara and Maugeri, 2019). Such a long record is useful to investigate precipitation changes, but station data need to be considered carefully, because the spatio-temporal heterogeneity of the data availability might be a cause of spurious trends (e.g. Masson and Frei, 2016). Interpolation of station data can be used to produce gridded products, commonly used to investigate climate variability. For example, SAFRAN (Durand et al., 2009) and SPAZM (Gottardi et al., 2009) are two reanalyses covering the French Alps, HISTALP (Auer et al., 2007) and EURO4M (Isotta et al., 2014) are a reconstruction of the climate of the European Alps, and E-OBS is a dataset commonly used for model verification over Europe (Cornes et al., 2018). General Circulation Models (GCMs) are also widely used to investigate climate change processes and trends. However, their coarse resolution precludes accurate simulations of small scale processes typical of mountainous areas, such as those inducing the spatial heterogeneity of precipitation and snow cover. It is therefore difficult to study the Alpine climate variability with GCMs (Zubler et al., 2015). Due to their finer resolution (~25 to 1 km) and their more detailed parametrization for physical processes (e.g. precipitation microphysics, surface snow scheme), that can be developed and tuned specifically considering regional geographical features, Regional Climate Models (RCMs) are more adapted to simulate the mountainous climate and can be used to dynamically downscale global climate model or global reanalyses. They have been widely used to simulate the climate in mountainous areas, and in particular temperature (Smiatek et al., 2009), daily and sub-daily precipitation events (Torma et al., 2014; Pieri et al., 2015), precipitation trends (Giorgi et al., 2016) and snow cover (Steger et al., 2013). A fine resolution is critical to simulate correctly



precipitation. Increasing the resolution from 0.44° to 0.11°, Fantini et al. (2018) demonstrated
that a RCM better captures the spatial pattern and the seasonal cycle of precipitation as well as
its daily intensities and statistics. The improvement of the model performances allows currently
to run RCMs at high resolution, typically 1 to 5 km, in non-hydrostatic configurations, resolving
the convective processes, and allowing an accurate description of the snow cover in mountainous
areas (e.g. Ban et al., 2014, 2015; Luthi et al., 2019). Such configuration is however challenging
to run over more than one to two decades due to computational costs.
Internal climate variability in the North Atlantic basin induces large seasonal to multi-decadal
variability of precipitation over Europe (Dell'Aquila et al., 2018). Using gridded observations,
Efthymiadis et al. (2007) and Brunetti et al. (2006) described the spatiotemporal distribution of
precipitation in the European Alps, highlighting higher interannual variability in the Southern
compared to the Northern Alpine areas. Seasonally, higher interannual variability in winter and
autumn compared to other seasons has also been observed in the European Alps (Brunetti et al,
2009; Bartolini et al, 2009). At daily to interannual time scales, the variability of precipitation
across the Alps is influenced by large-scale atmospheric circulations and particularly by the
North Atlantic Oscillation (NAO). A general picture is that a positive phase of the NAO induces
higher precipitation rates over the Northern Alps and lower ones over the Southern Alps, and
vice versa for negative phases of the NAO. This general view nonetheless hides important
heterogeneities, as local conditions can drive strong variations of local temperature, wind and
moisture transport that may amplify or mitigate synoptical atmospheric patterns (Keiler et al.,
2010). In addition, the NAO-precipitation relationship over the Alps is not stationary over the
last two centuries (Brunetti et al., 2006). At decadal timescales, the Atlantic Multidecadal
Variability (AMV) affects the precipitation rates over the Alps as suggested by Brugnara and
Maugeri (2019) using correlations with station data. However, the compensating effects of the
thermodynamical and dynamical signals of the AMV on the surrounding continents might induce
a strong nonlinearity between the AMV and the precipitation rates in Europe (Qasmi et al.,
111  2017).

Over the last decades, precipitation rates have been declining for any period of the year in the
mediterranean area (Giorgi et al., 2008; Mariotti et al., 2015), a signal partly attributed to
anthropogenic forcings (Hoerling et al., 2012) and associated with a significant surface drying
(Douville et al., 2017). Precipitation trends during the last century are more contrasted over the
Alps, with a decrease in the Southeastern Alps consistent with the drying of the mediterranean
area (Schmidli et al., 2002; Auer et al., 2007; Brugnara and Maugeri, 2019) and an increase in
the annual precipitation in the North Western Alps, mainly driven by a positive and significant
trend in winter and at high elevation (Masson and Frei, 2014; 2016). Napoli et al. (2019)
suggested that the contrasted trends of precipitation between high mountains and lowlands over
the last decades is explained by the aerosol forcing that cools the atmosphere mainly at low
elevations since the air quality is better at high elevations. The contrasted seasonal trends in
precipitation needs to be considered separately since their sign can be different from one season
to another one. In addition, precipitation trends emerge from the natural variability only when





considering long time series, typically one century (Schmidli et al., 2002). This result is
consistent with previous studies claiming that precipitation rates did not show any significant
trend in the Alps when considering shorter periods, as found in Durand et al. (2009) in the
French Alps over the period 1958-2002.
Dry spells did not show any clear tendency over the last century in the Alps (Schmidli and Frei,
2005; Brugnara and Maugeri, 2019), whereas the length of dry episodes increased in the
Mediterranean area (Kuglitsch et al. 2010). On the other hand, Schmidli and Frei (2005)
highlighted contrasting seasonal trends of the wet spell duration over the Swiss Alps, with a
lengthening in winter and a shortening in summer. Using 104 rain gauge stations in Switzerland,
Schmidli and Frei (2005) found an increase of the high quantiles of precipitation by 10% to 30%
over 1901-2000, a result evidencing an increase of extreme precipitation in this area, that they
relate to global climate change. Considering the annual maximum of daily precipitation
(Rx1day), which is often used as an indicator of extreme precipitation, Brugnara and Maugeri
(2019) found a similar signal in Switzerland over 1890–2017, whereas they could not highlight a
clear trend of extreme precipitation in Northern Italy. Using a convection-resolving RCM at 2.2
km resolution over the European Alps, Ban et al., (2015) confirmed the significant decrease in
mean summer precipitation related to global warming, and also find an increase in extreme
precipitation events that asymptotically intensifies following the Clausius-Clapeyron
relationship. Using model data over the last 400 years and over the 21st century, Brönniman et
al. (2018) pointed out that heavy precipitation change should thermodynamically increase with
temperature increase at the daily timescale, following the Clausius-Clapeyron relationship.
However, they evidenced a limitation of the available moisture in the atmosphere at the seasonal
timescale, a signal that limits the increase of heavy precipitation during the summer, and induces
a shift of heavy precipitation events from late summer to early summer and early autumn.
RCMs are limited-area models that can be forced laterally by atmospheric reanalysis, often
available only for the last decades, such as the ECMWF reanalyses ERA40 (1958-2002, Uppala
et al., 2004), ERA-Interim (1979-2019, Dee et al., 2011) and ERA5 (1950 onward, Hersbach and
Dee, 2016). Recent reanalysis products available for the whole 20th century (e.g. ERA20C,
1900-2010; Poli et al., 2016) now offer the possibility to apply RCMs over longer periods
although such a reanalysis, assimilating few data, is less reliable than reanalysis built on the
satellite era. Most of the observational and modeling data used to investigate climate change in
the European Alps generally do not combine a daily resolution, a centennial availability and a
fine spatial resolution, precluding investigations of changes in mean and extreme precipitation.
This strongly limits the possibility to detect precipitation trends, especially when considering the
large internal variability that may overwhelm long-term trends. In this study, daily precipitation
variability and changes in the European Alps (referred to as the Alps in the rest of the text) are
investigated over the period 1903-2010, using observational datasets as well as the regional
climate MAR model applied with a resolution of 7 km and driven by the ERA20C reanalysis.
The material and methods, including a description of the model MAR and different observational
datasets are described in Section 2. The model outputs are compared with several observational





datasets in Section 3. Seasonal and annual trends of precipitation indices are estimated over
1903-2010 in section 4. A discussion associated to the conclusions is presented in Section 5.
**2. Material and methods**
**2.1 The MAR model**
In this section we provide a succinct description of the atmospheric regional model MAR
([http://mar.cnrs.fr/index.php](http://mar.cnrs.fr/index.php)), developed at ULiège (Belgium) and IGE (Grenoble). A more
detailed description can be found in Gallée and Schayes (1994), Gallée (1995) and Gallée et al.
(2001, 2005). MAR is a hydrostatic primitive equation model with a vertical coordinate defined
as sigma coordinates. The radiative transfer through the atmosphere uses the radiative scheme of
the ECMWF reanalysis (Morcrette, 2002). MAR includes a detailed scheme of clouds
microphysics with six prognostic equations for specific humidity, cloud droplets concentration,
cloud ice crystals (concentration and number), concentration of precipitating snow particles and
rain drops. The convective adjustment is parameterised according to Bechtold et al. (2001).
MAR is coupled to the one-dimensional land surface scheme SISVAT (Soil Ice Snow
Vegetation Atmosphere Transfer, De Ridder and Schayes, 1997; Gallée et al., 2001) that
includes a snow multi-layer scheme (Brun et al., 1992) including prognostic equations for
temperature, mass, water content and snow properties (dendricity, sphericity and size) as well as
an ice module (Lefebre et al., 2003). As pointed out in Messager et al. (2006), the physics of
MAR can be adjusted to the region of interest and can be used with relatively high resolutions
(40 to 7 km, finer resolutions would not be possible when using the hydrostatic configuration).
MAR is a limited area model that offers the big advantage to be able to be forced by most of
reanalyses (including ERA-20C) or CMIP5/CMIP6 global model outputs. MAR has been first
designed for polar regions (Gallée, 1995), i.e. Antarctica (e.g., Gallée et al., 1996; Naithani, J. et
al., 2002; Gallée et al., 2013, Amory et al. 2015; Agosta et al. 2019) and Greenland (Fettweis et
al., 2017). It has also been applied over tropical regions (Messager et al., 2004) to investigate the
precipitation variability (Gallée, 2004). MAR has also been used to simulate the climate at
midlatitudes (e.g. Wyard et al., 2016), and in particular to study changes of precipitation over
Europe (Doutreloup et al., 2019). It has been applied over various mountainous areas, e.g.
Himalaya (Ménégoz et al., 2013), Svalbard (Lang et al., 2015), Kerguelen archipelago (Favier et
al., 2016; Verfaillie et al., 2019) and Antarctic peninsula (Datta et al., 2017).
**2.2 MAR configuration**
We used the version 3.9.0 of the MAR model, which is an open source code available at
mar.cnrs.fr/. Here, the model has been applied over the European Alps with a domain that
extends from 43.5°N to 48°N and 5°E to 13°E and includes 140 x 90 grid points, at 7 km
horizontal resolution. This configuration is based on 24 levels in the atmosphere, 7 levels in the
soil, and the snow cover is described with a number of layers varying from 1 to 20. Here, MAR
is laterally forced with the 6-hourly outputs of the ERA-20C reanalysis and its dynamical and



physical schemes are applied with a time step of 60s. ERA-20C is one of the first reanalyses
available from 1900 to 2010, assimilating surface pressure and near-surface winds over the
ocean and forced by SST reconstructions (Poli et al., 2016). By assimilating a limited number of
variables corresponding to those available relatively homogeneously over 1900-2010, the
ERA20-C reanalysis is expected to be homogeneous over this period, with a limited risk of
including spurious trend. However this reanalysis is probably less accurate than other products
assimilating also humidity and temperature in the free troposphere, a weakness that could induce
model biases also in regional experiments as evidenced by Fettweis et al. (2017) over Greenland.
MAR has already been successfully tested with the ERA20-C reanalysis as boundary conditions
over Europe (Wyard et al., 2017; 2018). After a spin-up of 2 years, the model in general and the
surface component in particular are supposed to reach an equilibrium. Therefore, the first years
have been excluded from this analysis that hence focuses on the period 1903-2010.
**2.3 Observational datasets**
Observational data sets are used in this study for two objectives: first, a comparison between the
model outputs and the different precipitation datasets have been conducted over 1971-2008, a
period for which all the datasets are available (Section 3). Second, an analysis of the trends of
several precipitation indices is described using the MAR experiments and the MeteoSwiss
station data available since the beginning of the 20th century (Section 4). In addition to the
MeteoSwiss station data, the following datasets have been considered:
**HISTALP**
The HISTALP ("Historical instrumental climatological surface time series") is the first long-
term database of meteorological data collected in the European Alps, initiated in the 1990s (Auer
and Bohm, 1994) and completed later on (Auer, et al., 2007). This dataset is available for air
temperature at 2 meters, surface pressure, precipitation, sunshine duration and cloudiness
through two different configurations: a first one including monthly homogenized time series of
local measurements, with the first observations available from 1760 for temperature and 1800 for
precipitation and a second one provided as a 1° gridded version. The gridded version, used in our
study, is available from 1800 to 2014.
**EURO4M-APGD**
The Alpine Precipitation Grid Dataset (EURO4M-APGD, Isotta et al., 2014) is a 5kmx5km grid
analysis of daily precipitation, extending over the European Alps and covering the period 1971-
2008. It is based on rain-gauge networks, encompassing more than 8500 stations from Austria,
Croatia, France, Germany, Italy, Slovenia and Switzerland. This dataset was developed in the
framework of the European Reanalysis and Observations for Monitoring (EURO4M) initiative
and is simply named EURO4M thereafter.



**E-OBS**

E-OBS is provided by the ECA&D project under the Copernicus Climate Change Service (van den Besselaar et al., 2011). It is a daily gridded dataset over Europe for temperature and precipitation, produced from interpolation of local measurements at two available resolutions: 0.25° and 0.1° (the highest resolution is used in this study). It is available from 1950 and updated each year, and is commonly used for model evaluation (e.g. Guillod et al., 2017). The version 19 of E-OBS offers an estimate of the uncertainty related to the interpolation of local observations by providing a 100-member ensemble for each variable (Cornes et al., 2018).

**SAFRAN**

The SAFRAN reanalysis combines large scale reanalyses and forecasts with in-situ meteorological observations, including precipitation, in order to provide a reanalysis of meteorological conditions in the French Alps, at the scale of massifs with a typical surface area of 1000 km2 within which meteorological conditions are assumed to be homogenous but vary with elevation by steps of 300 m (Durand et al., 2009). Hence this reanalysis does not use a regular grid but focuses on the elevation depency of meteorological conditions.

**SPAZM**

The SPAZM dataset is a gridded product at a 1km resolution for minimum and maximum daily 2m temperature and daily precipitation in the French Alps, available over the period 1948-2009 and based on the interpolation of a dense network of rain gauges and temperature sensors from Météo-France and the French Company for Electricity (EDF; Gottardi et al., 2009). This dataset is provided with calibration parameters allowing to reproduce the stream flow observed in the Rhône valley with a hydrological model. These parameters are useful to correct the snowfall rates that are often underestimated in rain gauge measurements. The solid and liquid precipitation has been computed by considering 100% of solid precipitation below 0°C and 100% of liquid precipitation above 2°C, considering daily temperature. A linear relationship for the ratio liquid/solid precipitation has been applied between 0°C and 2°C. Following the calibration suggested by Picouet (2012), we applied a correction factor for solid precipitation of 1.5 in the Northern Alps and 1.3 elsewhere. The Northern Alps are defined here as the areas located north of Grenoble city (45.1885° N) and higher than 1000m asl. This apparently-arbitrary values have been developed and evaluated in the context of hydrological studies (Gouttevin et al., 2017).

**2.4 Statistical analysis and indices**

Mean and trends for the precipitation indices described in Table 1 are considered in this study, most of them selected from those recommended by the World Meteorological Organization (Peterson et al., 2001; http://etccdi.pacificclimate.org/list_27_indices.shtml). When comparing MAR outputs with observational datasets based on a higher resolution (EURO4M, SPAZM), these ones are linearly interpolated onto the MAR grid. The gridded data are based on regular grids, except the SAFRAN data that is provided with a georeferenced polygon describing the



location of each massif. Correlation and Root Mean Square Error (RMSE) are computed to
investigate the differences between datasets. The linear trends are computed with a linear least-
square regression and a two-sided p-value is computed to test if the trends are significantly
different from zero (Wald test with a t-distribution of the test statistic). Mean values and
normalized trends (in percentage of the averaged values) are computed annually and seasonally
for the precipitation rates and indices in the following sections. The seasons are computed by
averaging indices over December-January-February (DJF) for winter, March-April-May (MAM)
for spring, June-July-August (JJA) for summer, and September-October-November (SON) for
autumn.
**3. Climatology of the MAR experiment and the observational datasets**
**3.1 Spatial differences**
The Total Precipitation amount (TP) averaged over 1971-2008 simulated with MAR is shown in
Figure 1a, with annual mean ranging between 300 and 3000 mm. In a general way, TP is
stronger in the Northern than in the Southern Alps as expected due to the drier Mediterranean
climate in the South. A comparison between the MAR experiment and the EURO4M reanalysis
(Figure 1b) shows a good consistency between these two datasets in the lowlands (i.e. below
500m asl), with positive or negative differences barely exceeding 20% in these areas. Above
1500m, this difference is much higher with precipitation rates 40 to 80% higher in the MAR
experiment than in the EURO4M dataset. Such a difference is significant with respect to the
interannual standard deviation of precipitation that generally does not exceed 30% over the
domain of application of MAR (Figure A1). The comparison between MAR and SPAZM
(Figure 1c) shows similar differences in the lowlands as with EURO4M (~20%), but a better
agreement at high elevation, with differences barely exceeding 40%. Correlation pattern between
MAR and EURO4M is 0.59 and reaches 0.63 between MAR and SPAZM, suggesting a slightly
better spatial consistency between these last two products. The precipitation rates estimated from
SPAZM above 1500m asl are also higher in comparison with those provided by EURO4M
(Figure 1d, differences ranging between 20% to 80%). As snowfall rates in SPAZM have been
adjusted to fit the hydrological balance over the French Alps, this comparison suggests that the
high precipitation rates simulated with MAR above 1500m asl may be realistic. Overall, TP
simulated with MAR is relatively similar to TP estimated from SPAZM and show stronger
values with respect to EURO4M data over the Alps.
**3.2 Vertical gradients**
Vertical gradients of TP simulated with MAR are evidenced in Figure 2 (as yearly averages,
considering both wet and dry days), including also a comparison with the SAFRAN reanalysis
and the MeteoSwiss station data. In the MAR experiment, TP varies from 2 to 4 mm day-1 at
500m asl and from 3 to 5 mm day-1 at 2500m asl in the Southern Alps, and from 2 to 4 mm day-
1 at 500m asl and from 3 to 6 mm day-1 at 2500m asl in the Northern French Alps. The vertical
gradient estimated from the SAFRAN dataset is smaller than those simulated by MAR. The large
spread in the scatter plots of Figures 2a-b is explained by the spatial variability of the vertical
gradient of precipitation, which is stronger in MAR than in SAFRAN. Based on a comparison
with high resolution numerical weather forecast model and glacier mass balance in mountain
glaciers in the French Alps, it has been shown that SAFRAN most likely underestimates high
elevation winter precipitation (above 2000m asl), which lends credence to the higher gradients
simulated by MAR (Vionnet et al., 2019). MAR data are compared to the MeteoSwiss station
data in Figure 2c-d. A difference between the two datasets in terms of vertical gradients is found
both in winter (14% km-1 for the local observations vs 43% km-1 in the MAR data) and in
summer (12% km-1 for the local observations vs 20% km-1 in the MAR data). These seasonally
contrasted differences suggest a possible underestimation of the precipitation rates estimated
from rain gauge measurements in relation with snowfall undercatch issues (Kochendorfer et al.,
2017). However, the differences found in summer, i.e. a period with reduced snowfall rates, also
suggests an overestimation of the precipitation rates at high elevation in the MAR experiment.
Overall, it is difficult to accurately quantify the model biases at high elevation areas because of
the scarceness of the observational data above 2000m asl whereas a large number of the MAR
grid cells reaches an elevation ranging between 2000m asl and 3000m asl with a spatial
resolution of 7km (Figure 2c-d). The vertical gradient of precipitation simulated by MAR over
the entire domain of application reaches 38% km-1 (1.15 mm.day-1.km-1) in winter and 33%
km-1 (1.21 mm.day-1.km-1) in summer (Figure 2e-f). Overall, the spatial patterns shown in
Figure 2g-h for the MAR experiment and the MeteoSwiss station data  suggest that MAR is able
to reproduce the seasonal variations of precipitation rates, with large (small) values in summer in
the Northeastern (Southwestern) Alps and an opposite pattern in winter.
**3.3 Interannual variability and seasonal cycle**
The temporal evolution of precipitation and its seasonal cycle over the period 1971-2008 is
shown in Figure 3 for three sub-regions defined as the Southern Alps (SA, 43.5°N to 45.5°N and
5°E to 7.5°E), the North Western Alps (NWA, 45.5°N to 47°N and 5°E to 7.5°E) and the
Northeastern Alps (NEA, 45.5°N to 48°N and 7.5°E to 13°E), corresponding respectively to the
orange, blue and purple boxes in Figure 1a. These boxes have been defined according to the data
availability of the products used for comparisons as well as regional climatological conditions:
SA is largely affected by the Mediterranean dry conditions, NWA precipitations rates are mainly
related to Western low pressure systems, whereas NEA is more typical of a continental climate,
with wetter conditions in summer than in winter. Over 1971-2008, precipitation rates estimated
from the different datasets (EURO4M, SPAZM, E-OBS, HISTALP) show similar features. They
range between 2 and 4 mm day-1 in SA and take higher values in NWA and NEA, varying
between 3 and 5 mm day-1. Table 2 includes the interannual correlations between the different
datasets, that takes values above 0.8, except for HISTALP that correlates neither with EURO4M
nor with MAR. The correlations between MAR and the observational datasets range between
0.84 and 0.89, whereas the correlations among these different datasets systematically exceeds
0.9, even reaching 0.99 and 0.97 for EURO4M versus SPAZM in SA and NWA. Such high
correlation is probably due to the rain gauge network used to produce these gridded datasets that
is partially common between the two products. Overall these correlation values suggest a correct
interannual variability in the MAR experiment, but less realistic than those estimated from the





other observational datasets (except HISTALP). The model bias could be related to both model
deficiencies and uncertainties in the ERA20C reanalysis used as boundary conditions. The
difference between the MAR experiment and the observational products is similar in all the
datasets, with a RMSE ranging between 0.26 and 1.14. RSME computed with HISTALP are
comparable to the other ones, evidencing that HISTALP is able to provide an estimation of
precipitation similar to the other products, even if the interannual variability is poorly
reproduced in this dataset compared to the other ones (low correlation). No clear regional and
data-dependant specificity in terms of RMSE is discernable in Table 2. It is worth noting that the
uncertainty related to data interpolation in E-OBS (orange dashed lines) encompasses the time
series corresponding to the other datasets. Such a large uncertainty is probably related to the low
number of observations assimilated in E-OBS over the Alps in comparison with the large
number of meteorological stations used to build EURO4M and SPAZM. It also points out the
large uncertainty of such gridded products in mountainous areas. MAR is closer to SPAZM and
EURO4M (smaller RMSE) than to HISTALP and E-OBS (higher RMSE), a finding giving
confidence in the realism of the MAR simulation. The different datasets show specific seasonal
cycles of precipitation (Figure 3, bottom), with two maximum values in spring and autumn in SA
(Mediterranean climate), one maximum value in summer in NEA (continental climate) and a
mix between these two regimes over NWA. Overall, the consistency between the different
datasets is better in SA and NEA than over NWA, where more discrepancies among the datasets
have been found. The comparison between model experiments and observational datasets gives
confidence in the precipitation rates simulated with the MAR model forced here by ERA-20C.
Longer simulations are therefore considered in the next section to investigate potential trends
over the last century.

## 396    4. Precipitation trends in the Alps over 1903-2010

### 397    4.1 Mean annual and seasonal precipitation

No significant trend in precipitation rates is identified on average when considering the three
Alpine sub-regions shown in Figure 3, a finding that could be explained by the shortness of the
time series. The seasonal trends of mean precipitation (STP) are investigated now over a longer
period, by considering in Figure 4 the MAR experiment (shaded) and the MeteoSwiss station
data (dotted) over 1903-2010. The MeteoSwiss data is used here because it is available over the
period covered by the MAR experiment (i.e. from 1903). Trends are contrasted both seasonally
and altitudinally. In winter, there is a general positive trend in precipitation over the Alps (up to
40% per century), contrasting with a drying simulated in the surrounding plains (~10% per
century), both in the Po plain and in the Rhône valley (Figure 4a). In winter, the MAR
experiment is consistent with the local observations showing an increase in precipitation over
Switzerland, except in the southern part of this country, where some stations show a drying
trend. However, when masking the non significant trends (p-value>0.05, Figure 4c), only the
positive trends of precipitation remain over the Alpine mountains, with values ranging between
20 and 40% per century in the Northern French Alps and in the Southwestern Switzerland, at an
elevation generally higher than 1500m asl. The MeteoSwiss data shows a significant increase in





precipitation at stations located further north in Switzerland, with a magnitude similar to the
model values. In summer, a general drying is simulated over the whole model domain (Figure
4b), with values exceeding 40 to 50% per century in the Po and the Rhône valleys. This signal is
much less pronounced in the alpine areas located above 1500m asl, where the signal is generally
not significant (Figure 4d). The drying is less pronounced over Switzerland in comparison with
the Southern Alps. Considering Figure 4b, a drying and even a slight increase in precipitation
(<10% per century) is found in the station data over the mountains of Southern Switzerland,
whereas a drying is observed in the north of the country (10% to 20% per century). The summer
trends locally observed in Switzerland are however barely significant (Figure 4d). Spring and
autumn trends show intermediate patterns, with a drying pronounced in the plains, and a slight
moistening in the mountainous areas (Figure 4e-f). The trends simulated and observed during
spring and autumn are however smaller and barely significant, except in the Po plain where a
drying is simulated by MAR and in the North of Switzerland where a significant moistening is
significant at some stations (Figure 4g-h).

**4.2 Wet spells**

Changes of the wet spell features are highlighted in Figure 5, showing the centennial summer
and winter trends of WD (Figure 5a-b), SDII (Figure 5c-d), MNWS (Figure 5g-h) and MWSD
(Figure 5e-f). In winter and over the Alps, the slight increase in WD, ranging between 0 and
10% per century, is slightly contributing to the increase in the STP (Figure 5a), but this one is
mainly explained by an increase in SDII by 10 to 30% per century (Figure 5c). These changes
are associated with an increase in the MWSD by 10 to 20% per century (Figure 5e). Finally, the
winter trend in the MNWS shows a different pattern from the previous ones (Figure 5g), with a
consistent decrease over the Northern Alps and in particular over Switzerland, ranging between
10 and 20% per century in both model and observational data. These changes suggest that the
precipitation increase found in winter over the Alpine mountains is related to longer, more
intense and less frequent wet spells. The agreement between the model and the observations is
relatively good in winter. In summer, a large part of the drying is explained by a reduction of
WD, reaching 10 to 20% in the north-western flank of the Alps, and exceeding 40 to 50% per
century in all the Southeastern Alps (Figure 5b). The SDII shows smaller changes during this
season (Figure 5d), that can be either positive or negative and barely exceeding 10% per century.
A general decrease of MWSD is also found in summer (Figure 5f), especially pronounced over
the Southeastern flank of the Alps and in the Po plain, a signal consistent between the model and
the observations over Switzerland. Finally, the model shows a general increase in MNWS
(Figure 5h) in summer, which is particularly strong over the Southeastern flank of the Alps. Over
Switzerland, this signal is small and not consistent with the observations that show a minor
reduction of MNWS. Overall, these results suggest that the decrease of the mean precipitation
rates over the Alps in summer is explained by a dramatic reduction of WD, with shorter and
more frequent wet spells without any clear change of SDII.





### 4.3 Extreme precipitation

The means and the trends of extreme precipitation rates are described by considering Rx1day seasonally (Figure 6) and annually (Figure 7). Extreme precipitation shows contrasted seasonal and spatial climatological patterns (Figure 6a-c-e-g). In winter and spring, it takes higher values over the Southern Alps (both in France and in Italy) than in the Northern Alps (Figure 6a-c). In summer, extreme values are smaller in the Western Alps than in the Eastern Alps. The most intense events are found in autumn, in both the Western and the Eastern parts of the Southern Alps, with mean Rx1day ranging between 50 and 70 mm day-1 in many areas (Figure 6g). Over Switzerland, the climatological pattern of Rx1day is consistent between the model and the observations, except for some local exceptions. In contrast with the trends of mean precipitation (Figure 4), there is a general increase in Rx1day (Figure 6b-d-f-h) that occurs over the four seasons and both in the plains and in the mountainous areas. The trend in the seasonal Rx1day over the period 1903-2010 takes values between 0 and 40% per century, and generally ranges between 20% and 40% per century over the Alpine mountain range. Such increase in percentages, suggests strong changes of local extreme, in particular in the regions where strong events are frequent in autumn (50 to 70 mm day-1, Figure 6g). The magnitude of the change in Rx1day is similar between the model and the observations, but the MeteoSwiss station data show more heterogeneous spatial trends, with some locations where the trends can be small and even slightly negative. The mean increase in Rx1day over the model domain reaches 10, 10, 6 and 13% per century, respectively in winter, spring, summer and fall, confirming a constant increase in Rx1day for all the seasons.

The climatological mean of the annual Rx1day (Figure 7a) is dominated by the seasonal Rx1day computed in autumn that shows a similar pattern (Figure 6g). Centennial Rx1day (Figure 7b, maximum Rx1day computed over 1903-2010) shows a pattern similar to the annual average of Rx1day. The strongest centennial events are found in the Southern Alps, in France and Italy, as well as in the Ticino valley in Switzerland, with extreme values locally reaching 250 to 300 mm day-1. The annual trends are inspected in Figure 7c, highlighting a general increase in extreme precipitation. The significance of this trend is described in Figure 7d, where non-significant trends (p-value>0.05) are masked. The annual samples considered here are obviously larger than the seasonal samples, allowing to detect more efficiently significant trends. In that case, a good consistency is found between the model and the MeteoSwiss station, both of them showing only positive trends, ranging between 20 and 40% per century, and covering large parts of the Alps and the Apennines.

### 4.4 Statistical significance of the trends

The large interannual to decadal variability of the climate in the Europe-North-Atlantic region, in relation with internal climate variability, makes precipitation trends challenging to detect in the Alps. When considering time series over 1971-2008, i.e. a relatively short period, there is no clear tendency of the mean precipitation over the Alps (Figure 3). By computing trends from the beginning of the 20th century, winter increase and summer decrease of mean precipitation

emerge from interannual to decadal variability over large areas (Figure 4 c-d). Similarly, the
positive trends of the annual Rx1day are significant (p-value<0.05) over large areas of the Alps,
in particular where the trends are positive and strong (i.e. >10% per century), whereas all the
negative and small trends in both the model data at the MeteoSwiss station data are not
significant (Figure 7d). Figure 7e illustrates the trend of the annual Rx1day, considering the
average of the MeteoSwiss data (purple), the model data integrated over Switzerland (blue) and
the model data integrated over the whole model domain (red). In agreement with the patterns
shown in Figure 7c-d, the mean trend in the annual Rx1day in Figure 7 is positive for both the
model outputs and the observations, reaching 9.43, 8.74 and 8.40 mm day-1 per century
respectively for the MeteoSwiss data, the model over Switzerland and the model over the whole
domain. The vertical bars in Figure 7e show the length of the time series required to identify a
significant trend over the period 1903-2010, starting from 2010 and going back into the past.
Time series has to be considered from 1961, 1932 and 1942 until 2010 to get significant trends
(p-value<0.05) for the MeteoSwiss data, the model integrated over Switzerland and the model
integrated over the whole domain respectively, which correspond to length of ~50 to ~80 years.
We nonetheless see that shorter time series can lead to significant signals as evidenced in Figure
7f showing the p-value computed to estimate the significant level of the trend. The three curves
reach values below 0.05 during the years 1960-1970, highlighting significant trends of Rx1day
over 1960-2010 for the observations and over 1970-2010 for the model data.
**5. Discussions and conclusions**
Previous work highlighted a temperature increase in the Alps over the last century, associated
with an increase in precipitation over the Northern Alps in winter and a drying in summer
(Schmidli et al., 2002; Auer et al., 2007; Masson and Frei, 2014; 2016; Brugnara and Maugeri,
2019). The detailed features of climate change over the Alps remain however partly unknown, in
particular because of the lack of observational data at high elevation, and observational issues
with snow precipitation. In addition, the large internal variability of the climate system,
especially pronounced in Europe, may overwhelm long-term trends that remain challenging to
detect. Here, the MAR model has been used at a 7km resolution over the Alps, over the period
1900-2010 laterally forced with the ERA-20C reanalysis. A comparison with several datasets
(EURO4M, SPAZM, SAFRAN, E-OBS, HISTALP and local MeteoSwiss data) demonstrates
that the model is able to reproduce the climatological precipitation rates as well as both the
interannual variability and the seasonal cycle of precipitation over the Alps. The high resolution
used in the MAR experiment allowed a relatively correct representation of the topography, with
a large number of grid cells covering elevation between 2000m and 3500m, corresponding to
areas with few available observations. The vertical gradient of precipitation in the Alps is
estimated to 33% km-1 (1.21 mm.day-1.km-1) in summer and 38% km-1 (1.15 mm.day-1.km-1)
in winter, with values stronger in the Northern Alps than in the Southern Alps. The spatial
variability of this vertical gradient is large, being affected by the local climate conditions.
The model experiment confirms an increase in precipitation during the winter over the North
Western Alps, above 500m asl and significant and more pronounced above 1500m asl, with local





values of 20% to 30% per century over the period 1903-2010. This increase in precipitation is
related to stratiform precipitation, mainly explained by more intense precipitation during wet
days, and it is associated to longer albeit less frequent wet spells, a result consistent with
Schmidli and Frei (2005) over Switzerland. The model reproduces the general drying that
occurred in summer during the same period, exceeding 40% to 50% per century in the lowlands,
whereas it is much smaller (<10% per century) and not significant above 1500m asl. This drying
is not related to a decrease of the intensity of precipitation, but it is explained by a dramatic
reduction of the number of wet days, with shorter and more frequent wet spells. It is consistent
with previous work suggesting that the Alpine region appears as an exception in a drying region,
where local convective precipitation increases locally, in relation to surface warming and
sufficient moisture available in the soil at high-elevation areas (Giorgi et al., 2016).
The model reproduces the observed climatological patterns of the seasonal maximum
precipitation, and in particular the large values occurring in the Southern Alps in autumn,
exceeding 50 and even 70 mm day-1 on average over the period 1903-2010. Centennial events
reaching 250 to 300 mm day-1 are simulated in the Southern Alps, both in France and in Italy, as
well as in the Ticino valley. The seasonal maximum of precipitation shows a general increase
over this period and present during the four seasons, reaching values between 20% and 40% per
century in the Alps. An increase of strong precipitation has been evidenced in previous studies
over the European Alps in relation with higher temperature and moisture rates in the atmosphere
following the Clausius-Clapeyron relationship (e.g. Ban et al., 2015). However, this increase
might be modulated by the availability of moisture in the low troposphere and at the surface. The
drying of the surface may act as a positive feedback enhancing warming signals, that has been
identified as one of the causes of the Mediterreanean amplification (Brogli et al., 2019).
Bronnimann et al. (2018) suggested a limited increase in Rx1day during the summer in the Alps
due to moisture limitation, pointing out a shift of this index typically observed at the end of the
summer toward early summer and early autumn. Here, the MAR experiment shows an increase
of Rx1day both annually and for all seasons, suggesting sufficient moisture available at the
surface and in the low troposphere over 1903-2010 in the model experiment. This difference
with Bronnimann et al. (2018) could be explained by a relatively high resolution (7km) in our
model configuration as well as specific physical parameterisations. Nevertheless, the MAR
experiment evidenced a significant increase in the Rx1day only in mountainous areas whereas no
significant signal is not found in the lowlands. This result is consistent with Brugnara et al.
(2019) who observed an increase in the Rx1day in the mountain areas of Switzerland, where
enough moisture would be available, whereas the drying occurring in the Southern flank of the
Italian Alps may prevent any increase in Rx1day.
In the MAR experiments, the trends of the seasonal maxima of precipitation integrated over the
Alps are significant (p-value<0.05) only when considering long time series, typically 50 to 80
years depending on the area considered. Some of these trends are nonetheless significant when
computed over recent decades, from the 1960s/1970s to the 2000s. This suggests a recent
acceleration of the increase in extreme precipitation, whereas earlier periods with strong





precipitation also occurred, in particular during the 1950s/1960s. These ones could be explained
by internal climate variability and/or the non-linear response of the climate system to
anthropogenic greenhouse gases and aerosols. In particular, the cooling related to aerosol
forcing, which peaked during the 1970s/1980s (Koch et al., 2011) over Europe could have
masked the warming related to greenhouse gases, and temporarily prevented changes of extreme
precipitation. Further model investigations should be conducted to disentangle the variability of
the Alpine climate related to internal variability and external forcings. This research is needed to
anticipate possibly strong precipitation changes in the Alps under the accelerating global climate
change.

**Data availability**:
● MAR is accessible at https://gitlab.com/Mar-Group, and documented at http://mar.cnrs.fr/
● The outputs of the MAR experiements will be uploaded to a public server (EUDAT) if the
paper is accepted and before publication.
● The scripts used in this study will be put on github if the paper is accepted.
● Alpine Precipitation Grid Dataset (EURO4M-APGD), Version 1, is available on-line at
DOI**:**10.18751/Climate/Griddata/APGD/1.0
● ECAD can be downloaded at https://www.ecad.eu//download/ensembles/ensembles.php
● SAFRAN data are available upon request to s2m.reanalysis@meteo.fr. Work is in progress to
release the data set on a public repository.
● The SPAZM dataset has been provided by EDF and Météo-France for this research. It could
be made available to other researchers under a specific research agreement. Requests should
be sent to dtg-demandedonnees-hydro@edf.fr"

**Author contributions**: All authors contributed to design the study. XF, MM and JB ran MAR
experiments and produced the figures. MM wrote the manuscript and other authors contributed
with suggested changes and comments.

**Competing interests**: The authors declare no competing interests.

**Acknowledgements**: As part of the project CDP TRAJECTORIES, this work is funded by the
French National Research Agency in the framework of the " Investissements d'avenir" program
(ANR-15-IDEX-02). The authors thank the ERASMUS+ program and the VATEX project
founded under the program Labex OSUG@2020 (ANR10 LABX56). The author thank the
providers of observational datasets: MeteoSwiss, the federal office for meteorology and
climatology, for providing the data station over Switzerland as well as the EURO4M-APGD
dataset; the HISTALP group (http://www.zamg.ac.at/histalp); Électricité De France (EDF) for
providing the SPAZM dataset; the E-OBS dataset from the EU-FP6 project UERRA
(http://www.uerra.eu) and the data providers in the ECA&D project (https://www.ecad.eu). The
authors thank the "Institut du Développement et des ressources en Informatique" (IDRIS,
CNRS) and the GRICAD project (https://gricad.univ-grenoble-alpes.fr/) for providing computer
time for the simulations presented in this paper. The Figures have been produced with the python
package basemap (https://matplotlib.org/basemap/).





**Tables:**

| Index | Unit | Description |
|---|---|---|
| TP | mm year-1 | Total precipitation amount per year |
| STP | mm day-1 | Total precipitation amount, including wet and dry days, seasonal or annual |
| WD | days | Number of wet days (≥1mm) |
| SDII | mm day-1 | Simple daily intensity (TP/WD) |
| Rx1day | mm day-1 | Maximum daily precipitation, seasonal or annual |
| MNWS | no unit | Mean number of wet spells per season |
| MWSD | days | Mean wet spell duration, averaged over a season (WD/MNWS) |


Table 1: Annual and seasonal precipitation indices analysed in this study. Precipitation is the
sum of solid and liquid precipitation.

| | Southern Alps (SA) | Northwestern Alps (NWA) | Northeastern Alps (NEA) |
|---|---|---|---|
| **MAR - EURO4M** | **C=0.89; RMSE=0.46** | **C=0.88; RMSE=0.37** | **0.85; RMSE=0.41** |
| **MAR - SPAZM** | **C=0.87; RMSE=0.26** | **C=0.85; RMSE=0.45** | **-** |
| **MAR - E-OBS** | **C=0.84; RMSE=0.69** | **C=0.86; RMSE=0.83** | **C=0.79; RMSE=0.90** |
| **MAR - HISTALP** | **C=0.02; RMSE=0.64** | **C=-0.10; RMSE=0.73** | **C=-0.22; RMSE=0.66** |
| EURO4M - SPAZM | C=0.99; RMSE=0.38 | C=0.97; RMSE=0.62 | - |
| E-OBS - SPAZM | C=0.93; RMSE=0.63 | C=0.95; RMSE=1.14 | - |
| E-OBS - EURO4M | C=0.94; RMSE=0.29 | C=0.95; RMSE=0.54 | C=0.93; RMSE=0.54 |
| EURO4M - HISTALP | C=0.03; RMSE=0.68 | C=-0.08; RMSE=0.74 | C=-0.26; RMSE=0.67 |


Table 2: Correlation Pearson coefficient computed over 1971-2008 (C) and Root Mean Square
Error (RMSE) between the time series of precipitation data over 1971-2008 for the three
different subareas described in Figure 1. All the correlations are significant (p-value>0.1), except
those computed with the HISTALP data.





**Figure captions**

Figure 1: (a) Annual mean of precipitation (TC, mm year-1) over 1971-2008 in the Alps
simulated with the model MAR applied with a resolution of 7km and laterally forced with ERA-
20C. The colored boxes correspond to the Southern Alps (SA, orange), the Northwestern Alps
(NWA, blue) and the Northeastern Alps (NEA, purple). Precipitation differences (%) between:
(b) the MAR experiment and the SPAZM dataset, (c) the MAR experiment and the EURO4M-
APGD observational gridded datasets, and the SPAZM and the EURO4M-APGD datasets. (c)
and (d) are shown only for the area where the SPAZM data is available. The 1000 m-spaced
black contours show the topography in the 7km-resolution model, starting from 500 m.asl and
political frontiers are denoted with the black dashed lines. The pattern correlation between MAR
outputs and observational data is 0.59 and 0.63 respectively with EURO4M and SPAZM (p-
value<1e-200).

Figure 2: Precipitation (STP) averaged over 1971-2008 simulated by MAR and estimated from
reanalyses and observations. (a-b-c-d-e-f) show precipitation as a function of the elevation over
the Southern (a) and the Northern (b) French Alps in MAR experiments and SAFRAN
reanalysis, and estimated over Switzerland from local meteorological stations and from the MAR
grid cells covering Switzerland during summer (c) and winter (d). The vertical gradient averaged
over the full domain are shown for winter (e) and summer (f). STP averaged over 1903-2010 in
the MAR experiment (shaded, mm.day-1) and observed at MeteoSwiss stations in Switzerland
(dots, observations available from the beginning of the XXth century) are shown for summer (g)
and winter (h). In (g) and (h), the 1000 m-spaced black contours show the topography with a
7km-resolution, starting from 500 m.asl and political frontiers are denoted with the black dahsed
lines.

Figure 3: Annual mean precipitation (mm day-1) averaged over the Southern Alps (SA), the
Northwestern Alps (NWA) and the Northeastern Alps (NEA) over the period 1971-2008 (top
row) and corresponding monthly averaged seasonal cycle over the same period (bottom row).
The area covered by the SA, NWA and NEA domains can be visualized in Figure 1. MAR
outputs and observational data sets EURO4M, SPAZM, HISTALP and E-OBS are shown (see
text for details). E-OBS (orange solid line) is provided with an estimation of the observational
uncertainty (orange dashed lines).

Figure 4: Seasonal linear trends (percent per century) of precipitation over 1903-2010 in winter
(a-c), summer (b-d), spring (e) and autumn (f). 1000 m-spaced black contours show the
topography in the 7km-resolution model, starting from 500 m.asl and frontiers are denoted with
the black dashed lines. In (c-d) and (g-h), the trend is masked when its p-value is below 0.05
(level of confidence is lower than 95%; white areas for the model outputs and station data
excluded).





Figure 5: Winter and Summer 1903-2010 trends (percent per century) of WD (a-b), SDII (c-d),
MWSD (e-f) and MNWS (g-h) simulated by the MAR model (shaded) and locally observed in
Switzerland (dots). WD is computed as a percentage of the available daily data for observations.
Any station from the observational network including missing data is excluded when computing
the trends.
Figure 6: Seasonal mean (mm day-1, left) and trends (percent per century, right) over the period
1903-2010 of the seasonal Rx1day simulated by MAR (shaded) and locally observed in
Switzerland (dots) for Winter (a-b), Spring (c-d), Summer (e-f) and Autumn (g-h).
Figure 7: Mean (a) and maximum (b) of the annual Rx1day and its trend including (c) and
excluding (d) the areas where the p-value is lower than 0.05, over 1903-2010. Annual Rx1day
(e) and associated p-value of the trend (f) over the same period averaged over the model domain
(red for the entire Alpine region and blue for the Swiss domain only) and for the MeteoSwiss
network (purple, average of the MeteoSwiss data station available over 1903-2010). The vertical
bars in (e) highlight the year before which the p-value never exceeds 0.05.



**Figures:**

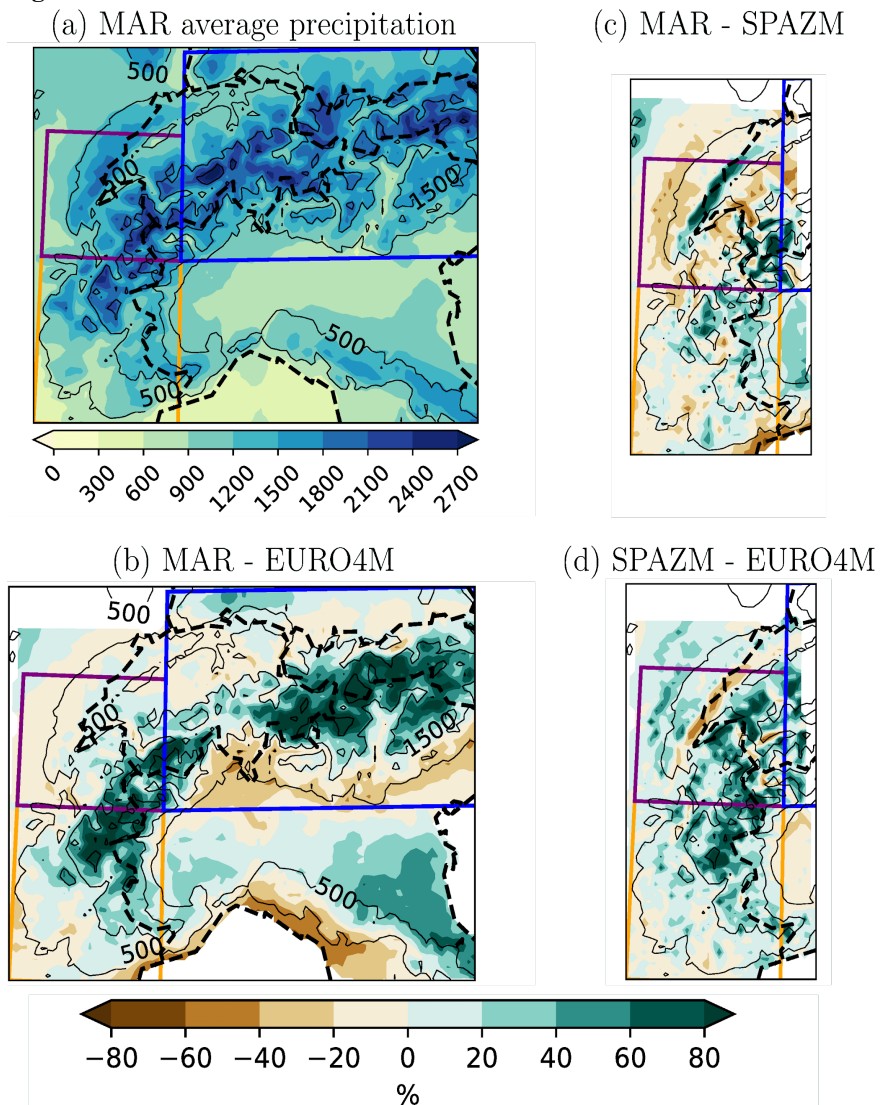


Figure 1: (a) Annual mean of precipitation (TC, mm year-1) over 1971-2008 in the Alps
simulated with the model MAR applied with a resolution of 7km and laterally forced with ERA-
20C. The colored boxes correspond to the Southern Alps (SA, orange), the Northwestern Alps
(NWA, blue) and the Northeastern Alps (NEA, purple). Precipitation differences (%) between:
(b) the MAR experiment and the SPAZM dataset, (c) the MAR experiment and the EURO4M-
APGD observational gridded datasets, and the SPAZM and the EURO4M-APGD datasets. (c)
and (d) are shown only for the area where the SPAZM data is available. The 1000 m-spaced
black contours show the topography in the 7km-resolution model, starting from 500 m.asl and
political frontiers are denoted with the black dashed lines. The pattern correlation between MAR
outputs and observational data is 0.59 and 0.63 respectively with EURO4M and SPAZM (p-
value<1e-200).





Hydrology and Earth System Sciences

Figure 2: Precipitation (STP) averaged over 1971-2008 simulated by MAR and estimated from reanalyses and observations. (a-b-c-d-e-f) show precipitation as a function of the elevation over the Southern (a) and the Northern (b) French Alps in MAR experiments and SAFRAN reanalysis, and estimated over Switzerland from local meteorological stations and from the MAR grid cells covering Switzerland during summer (c) and winter (d). The vertical gradient averaged over the full domain are shown for winter (e) and summer (f). STP averaged over 1903-2010 in the MAR experiment (shaded, mm.day-1) and observed at MeteoSwiss stations in Switzerland (dots, observations available from the beginning of the XXth century) are shown for summer (g) and winter (h). In (g) and (h), the 1000 m-spaced black contours show the topography with a 7km-resolution, starting from 500 m.asl and political frontiers are denoted with the black dahsed lines.

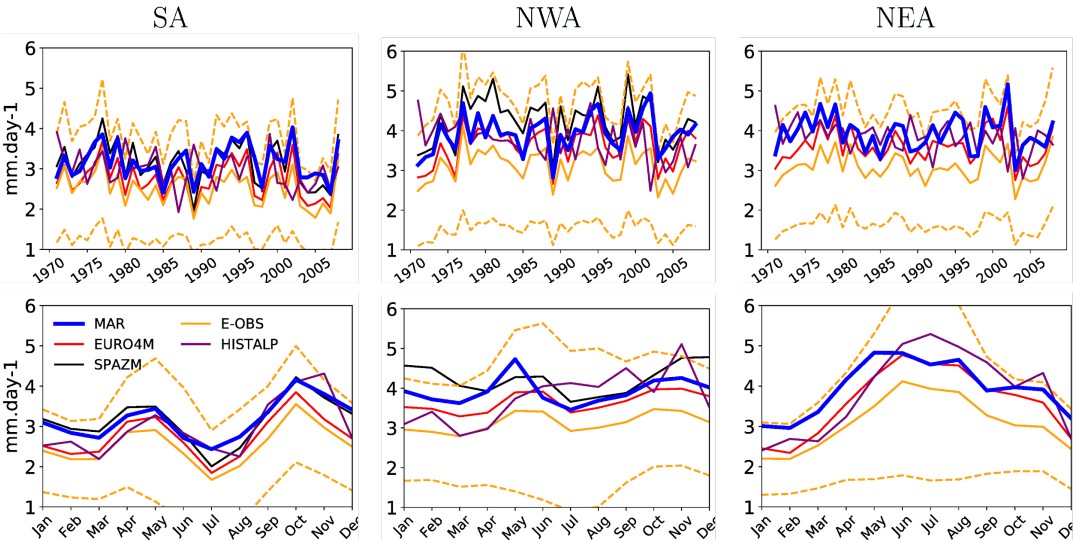

Figure 3: Annual mean precipitation (mm day-1) averaged over the Southern Alps (SA), the Northwestern Alps (NWA) and the Northeastern Alps (NEA) over the period 1971-2008 (top row) and corresponding monthly averaged seasonal cycle over the same period (bottom row). The area covered by the SA, NWA and NEA domains can be visualized in Figure 1. MAR outputs and observational data sets EURO4M, SPAZM, HISTALP and E-OBS are shown (see text for details). E-OBS (orange solid line) is provided with an estimation of the observational uncertainty (orange dashed lines).









Figure 4: Seasonal linear trends (percent per century) of precipitation over 1903-2010 in winter
(a-c), summer (b-d), spring (e) and autumn (f). 1000 m-spaced black contours show the
topography in the 7km-resolution model, starting from 500 m.asl and frontiers are denoted with
the black dashed lines. In (c-d) and (g-h), the trend is masked when its p-value is below 0.05
(level of confidence is lower than 95%; white areas for the model outputs and station data
excluded).









Figure 5: Winter and Summer 1903-2010 trends (percent per century) of WD (a-b), SDII (c-d),
MWSD (e-f) and MNWS (g-h) simulated by the MAR model (shaded) and locally observed in
Switzerland (dots). WD is computed as a percentage of the available daily data for observations.
Any station from the observational network including missing data is excluded when computing
the trends.









Figure 6: Seasonal mean (mm day-1, left) and trends (percent per century, right) over the period
1903-2010 of the seasonal Rx1day simulated by MAR (shaded) and locally observed in
Switzerland (dots) for Winter (a-b), Spring (c-d), Summer (e-f) and Autumn (g-h).






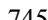




Figure 7: Mean (a) and maximum (b) of the annual Rx1day and its trend including (c) and excluding (d) the areas where the p-value is lower than 0.05, over 1903-2010. Annual Rx1day (e) and associated p-value of the trend (f) over the same period averaged over the model domain (red for the entire Alpine region and blue for the Swiss domain only) and for the MeteoSwiss network (purple, average of the MeteoSwiss data station available over 1903-2010). The vertical bars in (e) highlight the year before which the p-value never exceeds 0.05.



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





Appendix

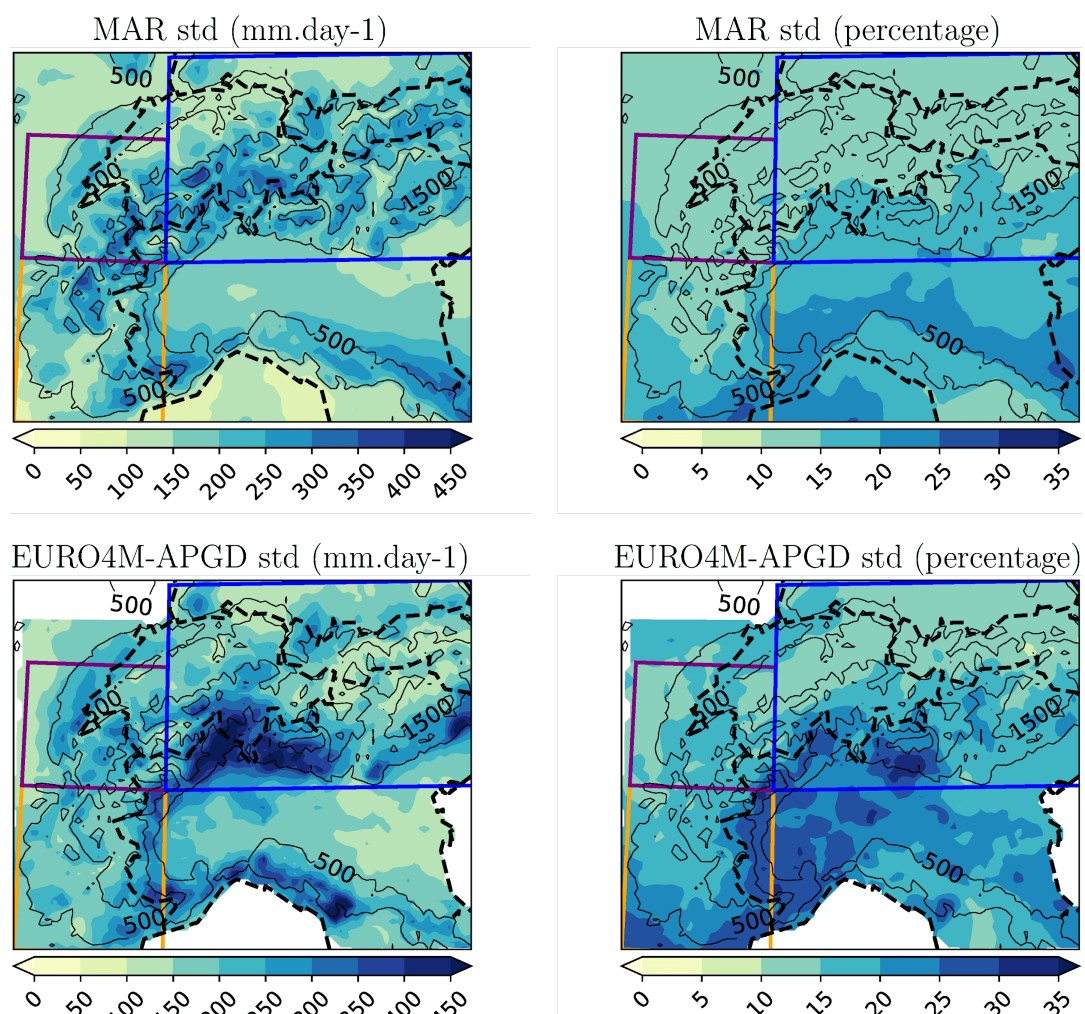

Figure A1: Standard deviation of annual precipitation over 1971-2008, in mm.day-1 (left)
and in percent (right), in the MAR experiment (top) and in the EURO4M-APGD dataset
(bottom).