# Peer review of "Contrasting seasonal changes in total and intense precipitation in the European Alps from 1903 to 2010"

_Hydrology and Earth System Sciences, 2019_

## Referee Comment (RC1) · Anonymous Referee #1 · 4 Mar 2020

Synopsis:

The paper analyses trends of precipitation in the Alpine region since 1900 in a dynamical downscaling simulation. They use the regional climate model MAR at 7 km resolution, driven with ERA-20C boundary conditions, for the Alpine region. The focus is is on extreme precipitation events and their seasonal changes. They first evaluate climatological features such as altitude gradients and then assess trends. They find seasonal, regional and altitude-dependent patterns of trends. Trends are mostly only significant when considering the entire centennial time period.

The paper is interesting and scientifically sound, though not ground-breaking. It fits into

the scope of the journal and hence I recommend publishing the paper after revisions have been taken into account.

Major points:

Structure: The paper could be shortened and made more accessible. For instance. The Introduction is long and talks at length about aspects such as the North Atlantic Oscillation etc. These aspects are not taken up later in the paper. I suggest to remove all parts that are not addressed later in the paper (or, alternatively, to revert to these aspects in the discussion part).

Trends: An important question regarding the trends reported is the trend in the driving data set, i.e. ERA-20C. Please report the trends in this data set. If this data set is drying out unrealistically or has other shortcomings, this might affect the final results and its interpretation.

Interpretation: It seems the authors interpret their findings mostly in terms of a regionally closed moisture budget (e.g. when they discuss different moisture availability as a function of altitutde), although they are not very clear about it. They do not address or discuss things like atmospheric rivers or the like. I think they should discuss their findings more broadly and more specifically. Also, perhaps they could give the reader some feeling about the change in moisture advection in ERA-20C or in MAR. Sometimes it is not clear whether they refer to a model diagnistic or their interpretation (e.g., when they write about moisture availability).

Minor:

L 24: longer and more intense?

L. 46: Danube: Alpine an non-alpine headwaters show quite different behaviour in some aspects. So, perhaps just at in brackets (Inn, Salzach, Saalach, etc.)

L. 50: make these

L. 54: has occurred and is expected to occur

L. 162: This is the first instance that a relation is made between the text and the paper. I suggest placing the paper in the context much earlier.

L, 165: I am missing what questions are actually addressed in the paper.

Sec. 2.3. For the sake of completeness, indicate the time resolution of all data sets (e.g. L. 239: is the gridded data set also monthly? Is SAFRAN daily?).

L. 310: 40-80% difference is huge!!

L. 315: Pattern correlation?

L. 319: Again, 20-80% is huge!

L. 366: I can hardly believe the bad correlation of HISTALP. Please double check.

L. 380: dependent

L. 426: twice significant

L. 449: Avoid qualifiers such as "dramatic"

L. 503: I do not understand this sentence.

Discussion: For trends in Rx1day I suggest the authors cite Scherrer et al. (2016), https://doi.org/10.1002/2015JD024634

L. 534: Just to clarify: stratiform precipitation is a separate diagnostic variable in the model?

L. 542: local convective precipitation locally?

---

## Referee Comment (RC2) · Federico Grazzini (Referee) · 10 Mar 2020

The manuscript by Ménégoz et al. investigates precipitation trends in the Alpine region through the aid of regional climate model MAR applied over the period 1903-2010. The model is forced by boundary condition from ERA-20C. Details of climate change over the Alps, especially for precipitations, need to be clarified further. Local feedbacks respect to air mass temperature increase, soil conditions, moisture availability also in relation with altitude in addition to changes in the dynamics of weather pattern are all factors potentially affecting precipitation distribution.

In that respect, regional climate models (RCM) are useful to enrich trends analysis and

overcome some limitations posed by availability, coverage and representativity of ob-servational network. One aspect emphasized in this work is the estimate of the vertical gradient of precipitation, hardly quantifiable over a region from sparse observations.

However, it is necessary to be very careful when comparing these results with obser-vations due to inevitable systematic errors introduced by modelling. Biases may be different for different precipitation types. For example, I would tend to believe that the variations of the winter stratiform precipitation are more solid than the trends deduced for warm seasons precipitation, due to the well-known difficulties in representing local convection, starting from the diurnal cycle up to the poor ability to simulate well convec-tive organization, especially when running at such intermediate resolution (7km). I think these limitations should be stated more clearly, especially in the introduction where a very optimistic idea is given about model ability to reproduce precipitation processes in such a complex geographical area.

Nevertheless, I found this valuable work and now I will focus on some specific issues to improve the manuscript.

1) In the introduction, where primary observational datasets are introduced, perhaps it worth to consider also the recent ARCIS dataset (https://www.arcis.it/wp/) which con-sists of a high-resolution climate precipitation analysis for north-central Italy (1961-2015). See also Pavan et al. 2019 for a description of the dataset which covers the entire south-alpine area at high-resolution.

2) At line 306 you state that the annual mean of your reconstructed climatology 1971-2008 is higher in the northern part of the Alps than in the south, and assume that is consistent with observations. This sounds like a wrong assumption since the annual precipitation maxima are recorded in the southern side of the Alps. Isotta et al. 2014 and Crespi et al. 2018, for example, is showing that annual maxima (EURO4M) are larger on the southern-side compared to the moist northern rim, where it rains more frequently but with less intensity. Piedmont-Ticino-Lombardy and Julian/Carnic Alps

and some part of the northern Apennine are well known to be hot spots of heavy precipitation in the Alpine area.

3) Your study area leaves out the eastern alpine region, which, as mentioned above, shows the annual maxima (the Carnic Alps). I found this choice a bit weird. Maybe justify this or reconsider in consideration of the title of the work from which one assumes all the Alpine range is considered

4) A precipitation bias on the northern side is evident in the MAR experiment, and this should be considered more trough the text.

5) The mean seasonal trends are not in total agreement with other works, especially over cold seasons. MAR positive trends are to be found in winter and Norther-Western Alps while in other seasons and regions the trend in the mean is mostly negative or neutral. From observational datasets (namely Isotta et al.2014, Pavan et al. 2019) we see a positive and significant trend in Autumn more than winter, mostly affecting regions with higher annual rainfall (south-side of eastern Alps and northern Apennine). This is also consistent with the findings of Brönnimann et al., 2018.

6) It would be interesting to discuss more the Autumn season in which precipitation in the Alpine area is more intense and often produced by an "optimal" synergy of synoptic-scale systems, which convey towards the region the necessary moist convergence, and mesoscale convective systems (Grazzini et al. 2019). In this respect, a RCM output may be useful to investigate the changes in the relative contributions of large-scale vs mesoscale systems to precipitations.

---

## Referee Comment (RC3) · Anonymous Referee #3 · 6 Apr 2020

In this manuscript the authors present a set of simulations to study if their model can reproduce the observed trends of precipitations over the alpine region during the last century. For that purpose, they set their regional model domain to run at a high resolution so that the orography and subsequent feedbacks can be more accurately represented. They then compare their trends with observations from a set of rain gauges over Switzerland and argue about which of these trends can be represented. I think the paper is of interest to the community and I recommend its publication after some revisions have been performed.

-Sections 4.1 and 4.2: The most characteristic pattern of your model is the drying on the Po valley over summer. This is most likely related to the evolution of convective processes in your model as the climate warms. The observations in the southern part of Switzerland do not capture this trend at all, therefore bringing doubts about the physical reliability of the simulated signal. I understand that it might be complicated to get station data over the Po valley, however some datasets like EOBS are open and provide data since 1950. You should compare at least if in these datasets there is also a similar signal to what the model predicts. In case that is not observed one should consider whether if the observed trends are physical or a simple artifact coming from the parameterization of convective processes in your model.

-Section 4.3: It would be a great addition to the paper to compare the trends observed in extreme precipitation with the Clausius-Clapeyron relation, or at least give an estimate on how much they change per degree of warming.

Small comments:

-L56 Name the reasons: snow cover feedback etc... -L100 Mention the typical timescales of the NAO -L183: 7km horizontal resolution on the gray zone of convection, is your parameterization prepared for running at those scales? If it is not scale dependent, did you test the behavior of the model when switching it off? At these scales, convection should appear already in a quite nice form, the use of a non-scale dependant parameterization might do more harm than good to the dynamics of the model. Perhaps in the future you should consider running a similar simulation turning off the parameterization of deep convection. I understand that this might be beyond the scope of this study, but you should mention that the model will likely be subject to some of the recursive biases of parameterized convection such as too frequent and too light precipitation spells. -L206: show the domain in a plot with the orography plotted and showing the size of your relaxation zone and the different analysis areas (SA, NWA, NEA). -L210: what is the resolution of ERA-20C? I think you should specify somewhere that the use of a regional model at such resolution is necessary to capture

the spatial heterogeneity of the orography. Otherwise people might wonder why did you not simply use the reanalysis for looking at the trends. -L337: It is interesting to note that the large amounts of precipitation measured during summer at mid altitudes ( ~6-8 mm/day; 500-1500m) cannot be predicted by the model. I guess these are likely stations strongly affected by convective precipitation, as the bias does not appear in winter. I wonder if this might happen due to including too much convective mixing in the atmosphere by your convective parameterization, therefore making precipitation to be too light where it should be much more stronger and intense. -L431: I think you mean Figure 4a? -L428-434: There are too many indexes used here that have not being presented before, some of them do not have names that make them easy to identify what they mean (SDII, STP, MNWS. . .), you should rewrite this part presenting the indexes before you analyze their trends. You should consider also a better naming for the indexes, SDII could just be Idaily , STP just Pseason etc. . . -L445: I do not agree that the model and the observations are consistent. The observations tend to show a very weak signal if any, while the model specially in the southern part shows a very negative signal that is not captured by the southernmost stations at all.

-L540: As I mention in the first comment one should check if the trend in the Po valley has been observed. This is a very important question. The climate projections from the EURO-CORDEX ensemble show a similar behavior for the future climate (decrease in mean precipitation explained by decrease in frequency). If this behavior has not being measured in the observations one would wonder about the reliability of these projections, which is indeed a very important result. This is critical as the use of a convective parameterization biases very strongly the precipitation frequency of the models, therefore it might be just the parameterization overreacting to a perturbation in temperature.

-Table 1: I like the idea of explaining all the indexes in a table, but I think the description of the indexes should also appear in the text at least the first time they are used.

[Figure]

690, 2020.

---

## Author Comment (AC1) · 6 Jul 2020

**Response to Anonymous Referee #1**

The referee comments are in italics, and the author responses have been written in blue

*Synopsis:The paper analyses trends of precipitation in the Alpine region since 1900 in a dynamical downscaling simulation. They use the regional climate model MAR at 7 km resolution, driven with ERA-20C boundary conditions, for the Alpine region. The focus is on extreme precipitation events and their seasonal changes. They first evaluate climatological features such as altitude gradients and then assess trends. They find seasonal, regional and altitude-dependent patterns of trends. Trends are mostly only significant when considering the entire centennial time period. The paper is interesting and scientifically sound, though not ground-breaking. It fits into the scope of the journal and hence I recommend publishing the paper after revisions have been taken into account.*

We acknowledge the referee #1 for his/her supporting comments. Her/his comments are answered below.

*Major points:*

*Structure: The paper could be shortened and made more accessible. For instance. The Introduction is long and talks at length about aspects such as the North Atlantic Oscillation etc. These aspects are not taken up later in the paper. I suggest to remove all parts that are not addressed later in the paper (or, alternatively, to revert to these aspects in the discussion part).*

The introduction has been shortened by around 20 lines, by excluding all the introductive parts that are out of the scope of the article. The conclusion has also been slightly shortened.

*Trends: An important question regarding the trends reported is the trend in the driving data set, i.e. ERA-20C. Please report the trends in this data set. If this data set is drying out unrealistically or has other shortcomings, this might affect the final results and its interpretation. Interpretation: It seems the authors interpret their findings mostly in terms of a regionally closed moisture budget (e.g. when they discuss different moisture availability as a function of altitude), although they are not very clear about it. They do not address or discuss things like atmospheric rivers or the like. I think they should discuss their findings more broadly and more specifically. Also, perhaps they could give the reader some feeling about the change in moisture advection in ERA-20C or in MAR. Sometimes it is not clear whether they refer to a model diagnostic or their interpretation (e.g.,when they write about moisture availability).*

We acknowledge the first referee for these useful comments. The analysis has been completed with a description of the circulation changes (wind and moisture) at the scale of Western Europe in ERA20-C. The climatology of the vertically integrated moisture and the vertically integrated moisture flux from ERA20-C is shown in Figure 1 from this response, highlighting that most of the moisture is related to Westerlies bringing moist air masses from the Atlantic in all seasons.

The presence of the mountains induces lower vertically integrated moisture in the Alps compared to surrounding areas. Overall, the vertically integrated moisture is stronger during warm areas and periods, in particular during the summer, when the warm atmosphere is able to contain large amounts of water vapour. A longitudinal gradient is found over the continental areas, with drier air masses in the Eastern continent during winter, spring and autumn (Figure 1-a-b-d), whereas large amounts of atmospheric moisture are found in Eastern Europe in summer (Figure 1c). Interpretations of the seasonal vertically integrated moisture in the atmosphere is delicate in the sense that atmospheric moisture is driven both by temperature and winds.

[Figure]

Figure 1: ERA20C climatology (1902-2010 average) of vertically integrated moisture (shading) and vertically integrated moisture flux (arrows). The rectangle highlights the domain of application used in this study. This figure has not been included in the revised manuscript.

Changes of atmospheric moisture are considered here for a better understanding of the precipitation changes over the Alpine area. The moisture trends (vertically integrated moisture and moisture fluxes) are shown in figure 2, a figure that has been included and described in the revised manuscript. The seasonal changes in precipitation simulated with MAR over the Alps (Figure 4 of the original manuscript, Figure 3 of this response) are strongly related to the

seasonal changes of moisture in the ERA-20C reanalysis used as boundary conditions (Figure 2 of this response). The drying in the Po Plain is related to a drying that occured over a large part of the Mediterranean area, in particular over the French and the Italian Mediterranean coasts, that propagated inland especially during the summer, when southwestern winds transport moisture at the East of the continent (figure 2c). During the winter and the autumn, the moistening over Germany, Benelux and the North of France and Switzerland is related to an increase of western moisture fluxes that bring moisture from the Atlantic, in particular over the Northern flank of the Alps. The coarse resolution of ERA-20C does not allow a fine estimation of the precipitation changes over the Alps, contrary to what can be done with the MAR experiments at higher resolution. Conversely, the spatial features of the precipitation changes over the Alps simulated by MAR are largely related to the large-scale moisture fluxes driven by the ERA20C forcing.

[Figure]

Figure 2: ERA20C linear trends over 1902-2010 of vertically integrated moisture (shading) and vertically integrated moisture flux (arrows). The rectangle highlights the domain of application used in this study. This figure is now Figure 5 of the revised manuscript.

[Figure]

Figure 3: Seasonal linear trends (percent per century) of precipitation amounts in Winter (a-b-c), Spring (d-e-f), Summer (g-h-i) and Autumn (j-k-l), for the time periods 1903-2010 for (a-b-d-e-g-h-j-k) and 1958-2010 for (c-f-i-l). 1000 m-spaced black contours show the topography in the 7km-resolution model, starting from 500 m.asl and frontiers are denoted with the black dashed lines. In subfigures (a-d-g-j), the trend is masked when its p-value is below 0.05 (level of confidence is lower than 95%; white areas for the model outputs and station data excluded). This figure has replaced Figure 4 of the initial/discussion manuscript.

In the revised manuscript, the model diagnostics have been shown separately from the interpretations, either related to local-scale or to large-scale changes found in the ERA-20C reanalysis. Also, caution has been considered in the manuscript in the interpretations shown in the conclusion, in particular for the points concerning the moisture availability and the convective processes.

*Minor:*
*L 24: longer and more intense?*
*L. 46: Danube: Alpine an non-alpine headwaters show quite different behaviour in some aspects. So, perhaps just at in brackets (Inn, Salzach, Saalach, etc.)*
*L. 50: make these*
*L. 54: has occurred and is expected to occur*
*L. 162: This is the first instance that a relation is made between the text and the paper. I suggest placing the paper in the context much earlier.*
*L, 165: I am missing what questions are actually addressed in the paper.*
*Sec. 2.3. For the sake of completeness, indicate the time resolution of all data sets (e.g. L. 239: is the gridded data set also monthly? Is SAFRAN daily?).*
*L. 310: 40-80% difference is huge!!*
*L. 315: Pattern correlation?*
*L. 319: Again, 20-80% is huge!*
*L. 366: I can hardly believe the bad correlation of HISTALP. Please double check.*
*L. 380: dependent*
*L. 426: twice significant*
*L. 449: Avoid qualifiers such as "dramatic"*
*L. 503: I do not understand this sentence.*

*Discussion: For trends in Rx1day I suggest the authors cite Scherrer et al. (2016), https://doi.org/10.1002/2015JD024634*
*L. 534: Just to clarify: stratiform precipitation is a separate diagnostic variable in the model?*
*L. 542: local convective precipitation locally?*

We have considered all the minor comments in the revised version of the manuscript. In particular, we have double checked our analysis based on HISTALP and we have included *Scherrer et al. (2016)* in our references. A deeper analysis concerning the changes in convective versus stratiform precipitation, two separate diagnostic variables in our model, is provided in the response to referee #2.

---

## Author Comment (AC2) · 6 Jul 2020

*Federico Grazzini (Referee)fgrazzini@arpae.it*

The referee comments are in italics, and the author responses have been written in blue

*The manuscript by Ménégoz et al. investigates precipitation trends in the Alpine region through the aid of regional climate model MAR applied over the period 1903-2010.The model is forced by boundary conditions from ERA-20C. Details of climate change over the Alps, especially for precipitations, need to be clarified further. Local feedbacks respect to air mass temperature increase, soil conditions, moisture availability also in relation with altitude in addition to changes in the dynamics of weather patterns are all factors potentially affecting precipitation distribution. In that respect, regional climate models (RCM) are useful to enrich trends analysis and overcome some limitations posed by availability, coverage and representativity of observational networks. One aspect emphasized in this work is the estimate of the vertical gradient of precipitation, hardly quantifiable over a region from sparse observations. However, it is necessary to be very careful when comparing these results with observations due to inevitable systematic errors introduced by modelling. Biases may be different for different precipitation types. For example, I would tend to believe that the variations of the winter stratiform precipitation are more solid than the trends deduced for warm seasons precipitation, due to the well-known difficulties in representing local convection, starting from the diurnal cycle up to the poor ability to simulate well convective organization, especially when running at such intermediate resolution (7km). I think these limitations should be stated more clearly, especially in the introduction where a very optimistic idea is given about model ability to reproduce precipitation processes in such a complex geographical area. Nevertheless, I found this valuable work and now I will focus on some specific issues to improve the manuscript.*

We acknowledge Federico Grazzini, acting as the referee #2, for his useful comments. To further investigate the details of climate change in the Alps, an analysis of atmospheric changes at the scale of Western Europe has been performed and included in the response to referee #1. This complementary analysis highlighted that the precipitation changes documented in our study are strongly related to the large scale moisture changes occurring over Europe and reproduced in the ERA-20C reanalysis that has been used as boundary conditions for the MAR experiment. A deeper analysis has also been carried out concerning the relationship between temperature and precipitation changes (see the response to referee #3), as well as a discussion on the convective processes in our model (included in this response). We fully agree that caution is required when studying convective processes with atmospheric models with such intermediate resolution (7 km) that is not adequate to correctly simulate local convective events. This has been underlined in the revised manuscript. Unfortunately, we cannot investigate the links between soil conditions and precipitation in our study because soil moisture is a variable that has not been saved in our model experiment. Finally, two separate periods have been considered to discuss precipitation changes, 1902-2010 and 1958-2010 (Figure 1 of this response, and updated Figure 4 in the revised manuscript). This allows a comparison with recent works focusing on the last decades and not over the last century. The comments are answered one by one below.

*1) In the introduction, where primary observational datasets are introduced, perhaps it worth to consider also the recent ARCIS dataset ([https://www.arcis.it/wp/](https://www.arcis.it/wp/)) which consists of a high-resolution climate precipitation analysis for north-central Italy (1961-2015). See also Pavan et al. 2019 for a description of the dataset which covers the entire south-alpine area at high-resolution.*

We acknowledge the referee for mentioning the ARCIS dataset (Pavan et al., 2019) that we did not know. A description of the precipitation trends for seasonal mean as well as other precipitation indices reported in this dataset has been included in the revised manuscript. The Figure 4 of our article that was describing the trends over 1902-2010 has been extended with additional panels showing the trends over the last decades (1958-2010). This allows a more direct comparison with the study based on the ARCIS dataset as well as other ones only available for the last decades and not for the whole century. This is important since the regional trends over the last decades differ from those diagnosed at the centennial scale. The main seasonal trends observed in the ARCIS network (see Pavan et al., 2019, Figure 8) are captured in the MAR model experiment (Figure 1 in this response), with a strong drying in the Po Plain during DJF, MAM and JJA that contrasts with precipitation increase in some mountainous areas during the same seasons. In Autumn (SON), the 1958-2010 pattern largely differs from those simulated over 1902-2010, without any clear drying over the Po Plain and a general precipitation increase, especially pronounced over the mountains. Our results are fully consistent with the results from Pavan et al. (2019) (Figure 1 of this response). The trend simulated and observed over 1903-1958 are shown in Figure 1 of this response, to explain how they can compensate signals estimated over the second half of the period. In particular, the winter increase in precipitation diagnosed over the whole period (1903-2010) is mainly related to a trend concerning the first half of the period (Figure 1a-b-c). The moistening occurring in autumn over 1958-2010 is partly compensated with the trends found over 1903-1958. Finally, the drying over the Po Plain is found almost for all the periods and the seasons.

[Figure]

Figure 1: Seasonal linear trends (percent per century) of precipitation in winter (a-b-c), spring (d-e-f), summer (g-h-i) and autumn (j-k-l), over 1903-1958 (a-b-d-e), over 1903-2010 for (g-h-j-k) and over 1958-2010 for (c-f-i-l). 1000 m-spaced black contours show the topography in the 7km-resolution model, starting from 500 m.asl and frontiers are denoted with the black dashed lines. This figure has been included in the revised manuscript, in place of Figure 4.

*2) At line 306 you state that the annual mean of your reconstructed climatology 1971-2008 is higher in the northern part of the Alps than in the south, and assume that is consistent with observations. This sounds like a wrong assumption since the annual precipitation maxima are recorded in the southern side of the Alps. Isotta et al. 2014 and Crespi et al. 2018, for example, is showing that annual maxima (EURO4M) are larger on the southern-side compared to the moist northern rim, where it rains more frequently but with less intensity. Piedmont-Ticino-Lombardy and Julian/Carnic Alps and some part of the northern Apennine are well known to be hot spots of heavy precipitation in the Alpine area.*

We partly agree with this comment. Precipitation rates are actually underestimated in the MAR experiment compared to the EURO4M reanalysis in the Piedmont-Ticino-Lombardy area and in some part of the northern Apennine, where strong precipitation annual mean are observed (Isotta et al., 2014). However, for the French side, the annual mean of precipitation rates are stronger in the North than in the South, both in the observations (Isotta et al., 2014) and in the model (Figure 1 of the manuscript). These seasonal features as well as the differences between model and observations have been detailed in the revised manuscript. In addition, there is a confusion in this comment between the annual mean and the mean of the annual maximum (mean of Rx1day over a long period). At L.306 of the discussion manuscript, only the annual mean of precipitation is discussed (and not the mean of the annual maximum). The mean of the annual maximum (mean of Rx1day) over the last decades is shown in figure 7a, where maximum values are actually stronger in the southern flanks of the Alps in both the model and the observations, as stated in the initial manuscript lines 473-484.

*3) Your study area leaves out the eastern alpine region, which, as mentioned above, shows the annual maxima (the Carnic Alps). I found this choice a bit weird. Maybe justify this or reconsider in consideration of the title of the work from which one assumes all the Alpine range is considered*

Unfortunately, we cannot cover the full alpine domain in our study, because the model domain is too small. This will be underlined in the manuscript, and a larger domain will be considered in further studies to allow more investigations covering the whole European Alps.

*4) A precipitation bias on the northern side is evident in the MAR experiment, and this should be considered more through the text.*

This bias has been discussed in detail in the initial manuscript (L304-323). As stated in the discussion article, the comparison between the MAR outputs and the EURO4M dataset suggests a strong overestimation of the annual mean precipitation in the Northern side of the Alps (Figure 1b of the manuscript). Over France, the SPAZM dataset, based on statistical interpolation of meteorological station data and calibrated to fit hydrological simulations at large basin scale, suggests higher precipitation rates at high elevation in comparison with EURO4M (Figure 1d of the manuscript). This suggests large uncertainties in observational datasets at high elevation, a point that limits the possibility to evaluate model outputs for these areas.

*5) The mean seasonal trends are not in total agreement with other works, especially over cold seasons. MAR positive trends are to be found in winter and Norther-Western Alps while in other seasons and regions the trend in the mean is mostly negative or neutral. From observational datasets (namely Isotta et al. 2014, Pavan et al. 2019) we see a positive and significant trend in Autumn more than winter, mostly affecting regions with higher annual rainfall (south-side of eastern Alps and northern Apennine). This is also consistent with the findings of Brönnimann et al., 2018.*

As stated previously, a comparison of the MAR outputs over the last decades and not over the whole century (Figure 1 in this response, included in the revised manuscript), shows a good agreement between the model and the observations (Swiss station data as shown in Figure 1 as well as the changes described in Isotta et al. 2014 and Pavan et al. 2019), in particular for the moistening observed in autumn and the drying observed in winter that is a feature only visible for the last decades and not for the last century.

*6) It would be interesting to discuss more the Autumn season in which precipitation in the Alpine area is more intense and often produced by an "optimal" synergy of synoptic-scale systems, which convey towards the region the necessary moist convergence, and mesoscale convective systems (Grazzini et al. 2019). In this respect, a RCM output may be useful to investigate the changes in the relative contributions of large-scale vs mesoscale systems to precipitations.*

The seasonal cycle of precipitation over the European Alps differs widely depending on the area considered. One maximum of precipitation is found in the Eastern part of the Alps in summer whereas small precipitation rates occur at the same time in both the South and the West of the Alps (Figure 3 of the manuscript and Figure 2a-b-c-d in this response). In these areas, a maximum of precipitation is found either in spring, autumn or winter depending on the area considered (Figure 2a-b-c-d in this response). The maximum values of Rx1day are indeed found in autumn (Figure 6g in the manuscript). These seasonal features are described in the manuscript.

Here, further investigations concerning the relative contributions of convective versus stratiform precipitations have been conducted (Figure 2 in this response). Stratiform precipitation is mainly related to large-scale atmospheric circulation and convective precipitation can be related either to local processes or to a mix between large-scale circulation and local processes, conditions favourable to extreme events of precipitations (Grazziani et al., 2020). The convective precipitation represents a significant part of total precipitation in summer, from 10% to 50% at high elevation areas and even stronger ratio in the plain, with values varying between 60% and 90% in the model experiment (Figure 2g in this response). This ratio is smaller than 10% in winter and takes intermediate values in spring and autumn from 10% to 40% (Figure 2e-f-h in this response). The trend of the convective to total precipitation ratio is shown over 1902-2010 (Figure 2i-j-k-l) and 1958-2010 (Figure 2m-n-o-p). Over the last century, a significant negative trend from 5% to 20% of this ratio is simulated everywhere except at high elevation during all the seasons but the winter. In winter, a slight increase (<5%) is found, but it is generally not significant (p-value>0.05). Similar winter trend is found over 1958-2010, whereas the trend

widely differs depending on the period considered for the three other seasons. Over 1958-2010, a strong increase in the convective to total precipitation ratio is simulated during the summer in large areas of the Alps (figure 2o), and to a lesser extent during the spring (figure 2n), whereas this signal is smaller in autumn (figure 2p). However, the signals simulated over 1958-2010 are not significant (p-value>0.05).

The interpretation of these contradictory signals (1903-2010 versus 1958-2010) is not straightforward, because convective precipitation depends on several variables, including atmospheric moisture availability at different levels in the atmosphere, soil moisture and temperature of both the atmosphere and the surface. The significant decrease of the convective to total precipitation ratio over 1902-2010 might be induced by changes of atmospheric moisture, with, for example, increase of moisture transport at large-scale that would impact differently stratiform and convective precipitation. Anyway, the intensified warming that took place during the last decades in areas with sufficient soil moisture is generally expected to favour an increase of the convective processes, as found in our experiments over 1958-2010 and suggested for future scenarios in high elevation areas by Giorigi et al. (2016). Here, the resolution used in our experiment (7km) could also have induced artefacts because the convective processes might be partly resolved with such intermediate resolution, whereas the convective parameterisation is still switched on. Further tests should be conducted with various resolutions to get sufficient confidence when simulating the convective changes over the Alps. This approach would complement the tests of different convective scheme in MAR experiments recently done with the MAR model (Doutreloup et al., 2019). Here, the results concerning the convective to total precipitation ratio are published in this response, but they have not been included in the final version of the article.

References:
Grazzini, F., Craig, G.C., Keil, C., Antolini, G. and Pavan, V., 2020. Extreme precipitation events over northern Italy. Part I: A systematic classification with machine learning techniques. *Quarterly Journal of the Royal Meteorological Society*, *146*(726), pp.69-85.
Doutreloup, S.; Wyard, C.; Amory, C.; Kittel, C.; Erpicum, M.; Fettweis, X. Sensitivity to Convective Schemes on Precipitation Simulated by the Regional Climate Model MAR over Belgium (1987–2017). Atmosphere 2019, 10, 34.

[Figure]

Figure 2: Seasonal mean of precipitation (a-b-c-d) for station data (dotted) and model data (shading). Seasonal contribution of convective to total precipitation (e-f-g-h) and their linear trends per century over 1903-2010 (i-j-k-l) and over 1958-2010 (m-n-o-p). Non-significant trends (p-value>0.05) are hatched. 1000 m-spaced black contours show the topography in the 7km-resolution model, starting from 500 m.asl and frontiers are denoted with the black dashed lines. This figure has not been included in the revised version of the manuscript.

---

## Author Comment (AC3) · 6 Jul 2020

The referee comments are in italics, and the author responses have been written in blue

*In this manuscript the authors present a set of simulations to study if their model can reproduce the observed trends of precipitations over the alpine region during the last century. For that purpose, they set their regional model domain to run at a high resolution so that the orography and subsequent feedbacks can be more accurately represented. They then compare their trends with observations from a set of rain gauges over Switzerland and argue about which of these trends can be represented. I think the paper is of interest to the community and I recommend its publication after some revisions have been performed.*

We acknowledge the referee#3 for his/her encouraging general comments and we answer the point-by-point list of comments below.

*Sections 4.1 and 4.2: The most characteristic pattern of your model is the drying on the Po valley over summer. This is most likely related to the evolution of convective processes in your model as the climate warms. The observations in the southern part of  Switzerland  do  not capture  this  trend  at  all, therefore  bringing  doubts  about  the physical reliability of the simulated signal. I understand that it might be complicated to get station data over the Po valley, however some datasets like EOBS are open and provide data since 1950. You should compare at least if in these datasets there is also a similar signal to what the model predicts.  In case that is not observed one should consider whether the observed trends are physical or a simple artifact coming from the parameterization of convective processes in your model.*

This comment is fully in line with those of referee#2 who is suggesting to consider the ARCIS precipitation dataset, based on Italian station data interpolated to produce a gridded dataset available over 1961-2015 (Pavan et al., 2019). A description of the precipitation trends for seasonal mean as well as other precipitation indices reported in this dataset has been included in the manuscript. Additional panels has been included in the Figure 4 to show the trends over a shorter  period  (1958-2010;  see  the  response  to  referee#2).  This  allows  a  more  direct comparison with the study based on the ARCIS dataset as well as other ones only available for the last decades and not for the whole century. This is important since the regional trends over the last decades differ from those observed at the centennial scale. Interestingly, the main seasonal trends observed in the ARCIS network (Pavan et al., 2019; their Figure 8) are consistent with our model experiment (Figure 1 in the response to referee #2), with a strong drying in the Po Plain during winter, spring and summer that contrasts with precipitation increase in some mountainous areas during the same seasons. In autumn (SON), the 1958-2010 pattern widely differs from those simulated over 1902-2010, without any clear drying over the Po Plain and general precipitation increase, especially pronounced over the mountains. These signals are clear both in Pavan et al. (2019) and in our study (Figure 1 of the response to referee #2). Pavan et al. (2019) report a drying over large areas of the Po plain about 1 to 2 mm-day-1.year-1  in  summer  (their  figure  8),  which  corresponds  to  a  strong  drying  at  the

centennial timescale in areas where the seasonal precipitation rates ranges between 100 and 200 mm. This comparison gives confidence in the strong drying simulated with MAR over the Po plain.

*Section 4.3: It would be a great addition to the paper to compare the trends observed in extreme precipitation with the Clausius-Clapeyron relation, or at least give an estimate on how much they change per degree of warming.*

We acknowledge the referee #3 for this relevant comment. Our article is focusing mainly on precipitation changes, and not on temperature changes. However, we have considered further investigations using the temperature to question the relationship between these two variables. Over the MAR domain, the relationship between the averaged Rx1day anomaly and the annual temperature average anomaly is significant (p-value<0.05) and reaches a positive trend by 3.11% °C$^{-1}$ (Figure 1 in this response). This value is smaller than the Clausius-Clapeyron relationship that reaches in theory 6–7%.°C$^{-1}$ (Trenberth et al., 2003). It is also smaller than the value of 7.7%.°C$^{-1}$ reported by Scherrer et al. (2016) using meteorological stations available in Switzerland over the last century. This is now discussed in the revised manuscript. As described in the original manuscript, the increase in Rx1day intensity in the MAR simulation occurred during all the seasons (Figure 6 in the initial manuscript), even during the seasons and areas for which the seasonal mean of precipitation is decreasing (Figure 4 in the initial manuscript). However, due to internal variability, the Rx1day intensity shows a strong interannual variability (Figure 1 in this response). The centennial increase in Rx1day also shows a strong spatial variability, with values ranging between 0 and 40% (Figures 6 and 7 in the original manuscript).

[Figure]

Figure 1: Rx1day anomaly (deviation from mean, %) as a function of the temperature anomaly simulated on average over the domain of application of the MAR model for the period 1903-2010. Anomalies are computed as differences between the annual mean and the average over 1903-2010 for temperature and Rx1day intensity, the latter computed as percentages.

The local relationship between the trend of annual surface air temperature versus the trend of Rx1day intensity is further investigated in Figure 2a, for grid points lower (blue) and higher (red) than 500 m.asl. There is no clear dependency between these variables over 1902-2010, except maybe for a subgroup of the blue points (right part of the plot). Overall, strong Rx1day increases are simulated both under high and low warming levels. The same conclusion is found when comparing the Rx1day trend with the temperature trend during the Rx1day (Figure 2b). Even when focusing on the more recent decades (1958-2010), a period when a strong warming took place, ranging between 0.15°C to 0.5°C per decade in our experiment (Figure 2c in this response), there is no clear local dependency between the temperature trend and the Rx1day intensity.

[Figure]

Figure 2: Trend of Rx1day versus trend of temperature over 1903-2010 (a-b) and 1958-2010 (c-d), for annual mean temperature (a-c) and daily temperature during the Rx1day (b-d). It is not planned to include this figure in the revised manuscript.

A deeper analysis of the relationship between temperature and strong precipitation has been conducted below by superposing the areas where the increase of annual Rx1day is positive and significant (Figure 3a in this response) with other variables (hatched areas for Rx1day signal superposed to other variables in Figure 3b-c-d-e-f). The increase in Rx1day intensity is simulated both in areas with a strong warming (e.g. Apennines) or a moderate one (e.g. over the Alps at high elevation; figure 3b). The temperature change during the Rx1day is positive and strong in the Apennines in the Western and Northeastern parts of the Italian Alps (up to 4°C.century-1) whereas it shows smaller variations in the Northern flank of the Alps, with even negative trends over the Jura (up to -4°C.century, Figure 3c). Nevertheless, the increase in the Rx1day intensity projects well on the pattern correlation between the Rx1day and the annual temperature. These findings suggest that warmer temperatures favour strong precipitation events at the annual timescale, but the lack of correlation between the trends in Rx1day and the trends in temperature (Figure 1 in this response) demonstrates that other processes than temperature changes affect the Rx1day intensity. One of them is the shift of the seasonality of the occurrence of Rx1day. As shown in Figure 3e, the Rx1day occurs, on average over 1902-2010, from July over the Northern flank of the Alps (day 180 to 210) to August-September (day 210 to 270) over the Southern flank of the Alps. In the Jura, the increase in Rx1day intensity is associated with a Rx1day shift from the summer to the autumn (+30 to +60 days). This explains the small and even negative centennial trend (Figure 3c) of temperature during the Rx1day in this area. Conversely, the strong warming occurring in the Southeastern flank of the Alps and over the Appennine is not associated with any clear change of the seasonality of the Rx1day (Figure 3f). Over Switzerland, Brönnimann et al. (2018) also suggested a shift of the seasonality of the Rx1day. Here, even with a similar finding, caution is required with this assumption since the shifts described in figure 3f are not significant (p-value>0.05). Rx1day positive trends are also simulated in other areas with both limited warming and without any shift of the seasonality of the Rx1day occurrence, and in particular in the Alps at high elevation (Figure 3). This suggests that other processes are at play to drive increases in the Rx1day intensity. Further investigations are required to disentangle which of them could drive these changes, both in the atmosphere (e.g. moisture flux and convergence at different elevations) and at the surface (e.g. soil conditions including moisture availability).

[Figure]

Figure 3: Rx1day intensity trend (a) temperature trend (b), temperature trend during the Rx1day occurrence (c) and correlation between Rx1day and annual temperature (d). Mean (e) and trend (f) of the occurrence day of Rx1day over 1903-2010. In all the panels, the hatches highlight the areas where the Rx1day trend is positive and significant (p_value<0.05). Temperature increase and change of the convective versus total precipitation ratio are significant (p-value<0.05) everywhere, whereas the trend (f) of the occurrence day of Rx1 is not significant (p-value>0.05). This Figure has been included in the revised manuscript.

*Small comments:*

*-L56 Name the reasons: snow cover feedback etc…*

*-L100 Mention the typical timescales of the NAO*

*-L183: 7km horizontal resolution on the gray zone of convection, is your parameterization prepared for running at those scales? If it is not scale dependent, did you test the behavior of the model when switching it off? At these scales, convection should appear already in a quite nice form, the use of a non-scale dependant parameterization might do more harm than good to the dynamics of the model. Perhaps in the future you should consider running a similar simulation turning off the parameterization of deep convection. I understand that this might be beyond the scope of this study, but you should mention that the model will likely be subject to some of the recursive biases of parameterized convection such as too frequent and too light precipitation spells.*

*-L206: show the domain in a plot with the orography plotted and showing the size of your relaxation zone and the different analysis areas (SA, NWA, NEA).*

*-L210: what is the resolution of ERA-20C? I think you should specify somewhere that the use of a regional model at such resolution is necessary to capture the spatial heterogeneity of the orography. Otherwise people might wonder why did you not simply use the reanalysis for looking at the trends.*

*-L337: It is interesting to note that the large amounts of precipitation measured during summer at mid altitudes (~6-8 mm/day; 500-1500m) cannot be predicted by the model. I guess these are likely stations strongly affected by convective precipitation, as the bias does not appear in winter. I wonder if this might happen due to including too much convective mixing in the atmosphere by your convective parameterization, therefore making precipitation to be too light where it should be much more stronger and intense.*

*-L431: I think you mean Figure 4a?*

*-L428-434: There are too many indexes used here that have not being presented before, some of them do not have names that make them easy to identify what they mean (SDII, STP, MNWS...), you should rewrite this part presenting the indexes before you analyze their trends. You should consider also a better naming for the indexes, SDII could just be daily , STP just P season etc…*

*-L445: I do not agree that the model and the observations are consistent. The observations tend to show a very weak signal if any, while the model specially in the southern part shows a very negative signal that is not captured by the southernmost stations at all.*

*-L540: As I mention in the first comment one should check if the trend in the Po valley has been observed. This is a very important question. The climate projections from the EURO-CORDEX ensemble show a similar behavior for the future climate (decrease in mean precipitation explained by decrease in frequency). If this behavior has not being measured in the observations one would wonder about the reliability of these projections, which is indeed a very important result. This is critical as the use of a convective parameterization biases very strongly the precipitation frequency of the models, therefore it might be just the parameterization over-reacting to a perturbation in temperature.*

*-Table 1: I like the idea of explaining all the indexes in a table, but I think the description of the indexes should also appear in the text at least the first time they are used.*

We have considered all these minor comments to prepare a new version of the manuscript. We fully agree with the comments concerning the resolution used that falls in the "grey zone", for which convective processes are partly resolved by the model dynamics, whereas the convective parameterization is still active. Further studies using different resolutions, switching on-off the convection parameterisation, non-hydrostatic configurations are different options that should be considered in future studies based on regional climate model experiments (some of these tests are already available in Doutreloup et al., 2019). Concerning the drying in the Po plain, and as mentioned before, this signal has been reported in the observations described in Pavan et al. (2019), which give confidence in this signal simulate with RCMs. Finally, the indices considered in the study and detailed in Table 1 have been presented in the text of the revised manuscript (Section 2.4), these are typical names from the World Meteorological Organization, named as the ETCCDI indices (Peterson et al., 2001; http://etccdi.pacificclimate.org/list_27_indices.shtml).

Reference:
Doutreloup, S.; Wyard, C.; Amory, C.; Kittel, C.; Erpicum, M.; Fettweis, X. Sensitivity to Convective Schemes on Precipitation Simulated by the Regional Climate Model MAR over Belgium (1987–2017). Atmosphere 2019, 10, 34.

---

## Author Response (AR2)

Dear Editor,

You will find in the hess web interface a new version of our manuscript hess-2019-690, that includes all the minor modifications suggested by both the editor and the anonymous referee. Only one point was not considered in the revised manuscript: the lon/lat were not included in Figure 1, because we have estimated that the political borders and the contours showing the topography allow a quick and easy visualisation of the location of the domain considered in the study. This is a way to avoid overloading the figure with additional information that is not used in the text, and that would appear with a small size of the police, difficult to read.

Best regards,

Martin Ménégoz